# A Simple Solution for Offline Imitation from Observations and Examples with Possibly Incomplete Trajectories

**Kai Yan**     **Alexander G. Schwing**     **Yu-Xiong Wang**
University of Illinois Urbana-Champaign
{kaiyan3, aschwing, yxw}@illinois.edu
https://github.com/KaiYan289/TAILO

## Abstract

Offline imitation from observations aims to solve MDPs where only *task-specific* expert states and *task-agnostic* non-expert state-action pairs are available. Offline imitation is useful in real-world scenarios where arbitrary interactions are costly and expert actions are unavailable. The state-of-the-art 'DIstribution Correction Estimation' (DICE) methods minimize divergence of state occupancy between expert and learner policies and retrieve a policy with weighted behavior cloning; however, their results are unstable when learning from incomplete trajectories, due to a non-robust optimization in the dual domain. To address the issue, in this paper, we propose Trajectory-Aware Imitation Learning from Observations (TAILO). TAILO uses a discounted sum along the future trajectory as the weight for weighted behavior cloning. The terms for the sum are scaled by the output of a discriminator, which aims to identify expert states. Despite simplicity, TAILO works well if there exist trajectories or segments of expert behavior in the task-agnostic data, a common assumption in prior work. In experiments across multiple testbeds, we find TAILO to be more robust and effective, particularly with incomplete trajectories.

## 1 Introduction

In recent years, Reinforcement Learning (RL) has been remarkably successful on a variety of tasks, from games [57] and robot manipulation [66] to recommendation systems [9] and large language model fine-tuning [53]. However, RL often also suffers from the need for extensive interaction with the environment and missing compelling rewards in real-life applications [37].

To address this, Imitation Learning (IL), where an agent learns from demonstrations, is gaining popularity recently [20, 22, 27]. Offline imitation learning, such as behavior cloning (BC) [51], allows the agent to learn from existing experience without environment interaction and reward label, which is useful when wrong actions are costly. However, similar to RL, IL also suffers when limited data is available [52] – which is common as demonstrations of the target task need to be collected every time a new task is addressed. In this work, we consider a special but widely studied [31, 36, 60] case of expert data shortage in offline IL, i.e., *offline Learning from Observations (LfO)* [71]. In LfO the task-specific data only consists of a few expert trajectories, key frames, or even just the goal (the latter is also known as *example-based IL* [13]), and the dynamics must be mostly learned from *task-agnostic* data, i.e., demonstration from data not necessarily directly related to the target task. For example, sometimes the agent must learn from experts with different embodiment [41], where expert actions are not applicable.

In the field of offline LfO, researchers have explored action pseudo-labeling [33, 60], inverse RL [31, 61, 73], and similarity-based reward labeling [8, 55]; for example-based IL, the benchmark is

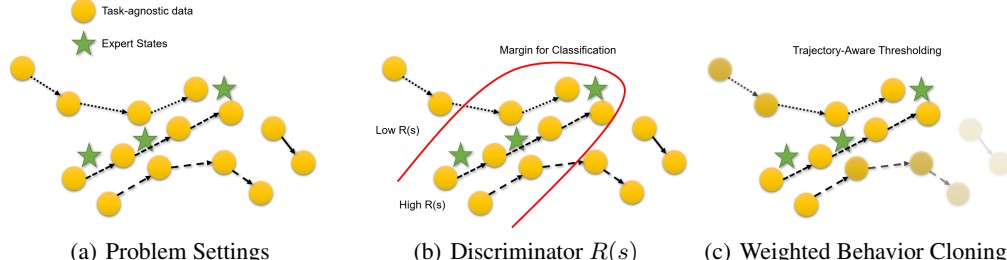

(a) Problem Settings  (b) Discriminator $R(s)$  (c) Weighted Behavior Cloning

Figure 1: An illustration of our method, TAILO. Different trajectories are illustrated by different styles of arrows. TAILO consists of two parametric steps: 1) train a discriminator which gives high $R(s)$ for near-expert states and low $R(s)$ for non-expert states, as shown in panel b); 2) conduct weighted behavior cloning with weights calculated from $R(s)$ along the trajectory, as shown in panel c). High transparency indicates a small weight for the state and its corresponding action.

RCE [13], which uses RL with classifier-labeled reward. Recently, DIstribution Correction Estimation (DICE) methods, LobsDICE [23] and SMODICE [41], achieve the state of the art for both offline LfO and example-based IL. Both methods minimize the state visitation frequency (occupancy) divergence between expert and learner policies, and conduct a convex optimization in the dual space.

However, DICE methods are neither robust to incomplete trajectories in the task-agnostic / task-specific data [72], nor do they excel if the task-agnostic data contains a very small portion of expert trajectories or only segments [56]. These are inherent shortcomings of the DICE-based methods, as they are *indirect*: they first perform optimization in a dual domain (equivalent to finding the value function in RL), and then recover the policy by weighted behavior cloning. Importantly, Kullback-Leibler(KL)-based optimization of dual variables requires complete trajectories in the task-agnostic data to balance all terms in the objective. Note, SMODICE with $\chi^2$-based optimization is also theoretically problematic (see Appendix C) and empirically struggles on testbeds [41].

To overcome the shortcomings listed above, we propose a simple but effective method for imitation learning from observations and examples, Trajectory-Aware Imitation Learning from Observations (TAILO). We leverage the common assumption that *there exist trajectories or long segments that are near-optimal to the task of interest in the task-agnostic data*. This assumption is the basis of skill-based learning [20, 46, 47], and the benchmarks of many recent works satiesfy this assumption [36, 40, 41, 56]; one real-life example fulfilling this assumption is robotics: the robot often utilizes overlapping skills such as moving the robotic arm and grabbing items from other tasks to complete the current task. Based on this assumption, we discriminate/identify which state-action pairs could be taken by the expert, and assign large weights for trajectory segments leading to those segments in the downstream Weighted Behavior Cloning (WBC). This is a simple way to make the learned policy *trajectory-aware*. The method only consists of two parametric steps: 1) train a discriminator using positive-unlabeled (PU) learning, and 2) use Weighted Behavior Cloning (WBC) on all state-action pairs in the task-agnostic data with the weights of WBC being a discounted sum over thresholded scores given by the discriminator. Note, the discounted sum propagates large scores to trajectory segments in the past far from expert states, if they lead to expert trajectory segments eventually. Meanwhile, as the task-agnostic data contains both expert and non-expert demonstrations, Positive-Unlabeled (PU) learning is better than plain binary classification. Fig. 1 summarizes our algorithm. We found this simple solution to be surprisingly effective across multiple testbeds, especially if the task-agnostic data contains incomplete trajectories. In this latter case, baselines struggle or even diverge. Moreover, we find our method to also improve if the task-agnostic data contains only a small portion of expert trajectories.

We summarize our contributions as follows: 1) We carefully analyzed the state-of-the-art DICE methods in offline LfO, pointing out their limitations both empirically and theoretically (see Appendix C for details); 2) We propose a simple yet effective solution to offline imitation learning from observations; and 3) We empirically show that this simple method is robust and works better than the state of the art on a variety of settings, including incomplete trajectories, few expert trajectories in the task-agnostic dataset, example-based IL and learning from mismatching dynamics.

## 2 Preliminaries

**Markov Decision Process.** A Markov Decision Process (MDP) is a well-established framework for sequential decision-making problems. An MDP is defined by the tuple $(S, A, T, r, \gamma)$, where $S$ is the state space and $A$ is the action space. For every timestep $t$ of the Markov process, a state $s_t \in S$ is given, and an action $a_t \in A$ is chosen by the agent according to its policy $\pi(a_t|s_t) \in \Delta(A)$, where $\Delta(A)$ is the probability simplex over $A$. Upon executing the action $a_t$, the MDP will transit to a new state $s_{t+1} \in S$ according to the transition probability $T(s_{t+1}|s_t, a_t)$ while the agent receives reward $r(s_t, a_t) \in \mathbb{R}$. The goal of the agent is to maximize the discounted reward $\sum_t \gamma^t r(s_t, a_t)$ with discount factor $\gamma \in [0, 1]$ over a complete run, which is called an episode. The state(-action pairs) collected through the run are called a state(-action) trajectory $\tau$. Trajectory segment in this work is defined as a continuous subsequence of a trajectory $\tau$. The state visitation frequency (*state occupancy*) of a policy $\pi$ is denoted as $d^\pi(s) = (1 - \gamma) \sum_t \gamma^t \Pr(s_t = s)$. See Appendix A for a detailed definition of state occupancy and other occupancies.

**Positive-Unlabeled Learning.** Positive-Unlabeled learning [12] addresses the problem of binary classification with feature $x \in \mathbb{R}^n$ and label $y \in \{0, 1\}$ when only the positive dataset $D_P$ and the unlabeled dataset $D_U$ are known. Our solution leverages *positive prior* $\eta_p = \Pr(y = 1)$ and *negative prior* $\eta_n = \Pr(y = 0)$, which are unknown and treated as a hyperparameter.

**Offline Imitation from Observations / Examples.** Offline imitation learning from observations requires the agent to learn a good policy from two sources of data: one is the *task-specific* dataset $D_{\text{TS}}$, which contains state trajectories $\tau_{\text{TS}} = \{s_1, s_2, \ldots, s_{n_1}\}$ from the expert that directly addresses the task of interest; the other is the *task-agnostic* dataset $D_{\text{TA}}$, which contains state-action trajectories $\tau_{\text{TA}} = \{(s_1, a_1), (s_2, a_2), \ldots, (s_{n_2}, a_{n_2})\}$ of unknown optimality to the task of interest. Note that the task-specific trajectory $\tau_{\text{TS}}$ can be incomplete; specifically, if only the last state exists as an example of success, then it is called *imitation from examples* [13, 41].

The state of the art methods in this field are SMODICE [41] and LobsDICE [23]. SMODICE minimizes the divergence between the state occupancy from task-specific data $d^{\text{TS}}$ and the learner policy $\pi$'s occupancy $d^\pi$; for example, when using a KL-divergence as the metric, the objective is

$$\min_\pi \text{KL}(d^\pi(s) \| d^{\text{TS}}(s)), \text{s.t. } \pi \text{ is a feasible policy.} \tag{1}$$

However, since the task-agnostic dataset is the only source of correspondence between state and action, the state occupancy of the task-agnostic dataset $d^{\text{TA}}$ must be introduced. With some derivations and relaxations, the objective is rewritten as

$$\max_\pi \mathbb{E}_{s \sim d^\pi} \log \frac{d^{\text{TS}}(s)}{d^{\text{TA}}(s)} - \text{KL}(d^\pi(s) \| d^{\text{TA}}(s)), \text{s.t. } \pi \text{ is a feasible policy.} \tag{2}$$

Here, the first term $R(s) = \log \frac{d^{\text{TS}}(s)}{d^{\text{TA}}(s)}$ is an indicator for the importance of the state; high $R(s)$ means that the expert often visits state $s$, and $s$ is a desirable state. Such $R(s)$ can be trained by a discriminator $c(s)$: a positive dataset $D_{\text{TS}}$ (label 1) and a negative dataset $D_{\text{TA}}$ (label 0) are used to find an 'optimal' discriminator $c = c^*$. Given this discriminator, we have $R(s) = \log \frac{c^*(s)}{1 - c^*(s)}$. SMODICE then converts the constrained Eq. (2) to its unconstrained Lagrangian dual form with dual variable $V(s)$, and optimizes the following objective (assuming KL-divergence as the metric):

$$\min_V (1 - \gamma) \mathbb{E}_{s \sim p_0}[V(s)] + \log \mathbb{E}_{(s, a, s') \sim D_{\text{TA}}} \exp[R(s) + \gamma V(s') - V(s)], \tag{3}$$

where $\gamma$ is the discount factor and $p_0$ is the distribution of the initial state in the MDP. In this formulation, $R(s)$ can be regarded as the *reward* function, while $V(s)$ is the value function. The whole objective is an optimization of a convex function with respect to the Bellman residual. With $V(s)$ learned, the policy is retrieved via weighted behavior cloning where the coefficient is determined by $V(s)$. LobsDICE is in spirit similar, but considers the occupancy of *adjacent state pairs* instead of a single state.

# 3 Methodology

## 3.1 Motivation and Overview

As mentioned in Sec. 2, the DICE methods for offline LfO discussed above consist of three parts: reward generation, optimization of the value function $V(s)$, and weighted behavior cloning. Such a pipeline can be unstable for two reasons. First, the method is *indirect*, as the weight for behavior cloning depends on the learned value function $V(s)$, which could be inaccurate if the task-agnostic dataset is noisy or is not very related to the task of interest. This is aggravated by the fact that $V(s')$ as a 1-sample estimation of $\mathbb{E}_{s'\sim p(s'|s,a)}V(s')$ and logsumexp are used in the objective, which further destabilizes learning. Second, for KL-based metrics, if no state appears twice in the task-agnostic data, which is common for high-dimensional environments, the derivative of the objective with respect to $V(s)$ is determined by the initial state term, the current state term $-V(s)$, and the next state term $\gamma V(s')$. Thus, if a non-initial state $s'$ is missing from the trajectory, then only $-\gamma V(s')$ remains, which makes the objective monotonic with respect to the unconstrained $V(s')$. Consequently, $V$ diverges for $s'$ (see Appendix C in the for a more detailed analysis and visualization in Fig. 7).

To address the stability issue, we develop Trajectory-Aware Imitation Learning from Observations (TAILO). TAILO also seeks to find an approximate reward $R(s)$ by leveraging the discriminator $c(s)$, which is empirically a good metric for optimality of the state. However, compared to DICE methods which determine the weight of behavior cloning via a dual program, we adopt a much simpler idea: find 'good' trajectory segments using the discounted sum of future $R(s)$ along the trajectory following state $s$, and encourage the agent to follow them in the downstream weighted BC. To do so, we assign a much larger weight, a thresholding result of the discounted sum, to the state-action pairs for 'good' segments. Meanwhile, small weights on other segments serve as a regularizer of pessimism [25]. Such a method is robust to missing steps in the trajectory. Empiricially we find that the weight need not be very accurate for TAILO to succeed. In the remainder of the section, we discuss the two steps of TAILO: 1) training a discriminator to obtain $R(s)$ (Sec. 3.2), and 2) thresholding over discounted sums of $R(s)$ along the trajectory (Sec. 3.3).

## 3.2 Positive-Unlabeled Discriminator

Following DICE, $R(s)$ is used as a metric for state optimality, and is obtained by training a discriminator $c(s)$. However, different from DICE which regards all unlabeled data as negative samples, we use Positive-Unlabeled (PU) learning to train $c(s)$, since there often are some expert trajectories or segments of useful trajectories in the task-agnostic dataset. Consequently, we treat the task-agnostic dataset as an *unlabeled* dataset with both positive samples (expert of the task of interest) and varied negative samples (non-expert).

Our training of $c(s)$ consists of two steps: 1) training another discriminator $c'(s)$ that identifies safe negative samples, and 2) formal training of $c(s)$. In the first step, to alleviate the issue of treating positive samples from $D_{\text{TA}}$ as negatives, we use a debiasing objective [29] for the training of $c'(s)$ (see Appendix A for a detailed derivation):

$$\min_{c'(s)} L_1(c') = \min_{c'(s)} -[\eta_p \mathbb{E}_{s \sim D_{\text{TS}}} \log c'(s) + \max(0, \mathbb{E}_{s \sim D_{\text{TA}}} \log(1 - c'(s)) - \eta_p \mathbb{E}_{s \sim D_{\text{TS}}} \log(1 - c'(s)))], \quad (4)$$

where the bias comes from viewing unlabeled samples as negative samples. Here, positive class prior $\eta_p$ is a hyperparameter; in experiments, we find results to not be sensitive to this hyperparameter.

In the second step, after $c'(s)$ is trained, $R'(s) = \log \frac{c'(s)}{1-c'(s)}$ is calculated for each state in the task-agnostic data, and the states in the (possibly incomplete) trajectories with the least $\beta_1 \in (0, 1)$ portion of average $R'(s)$ are identified as "safe" negative samples. Note, we do not identify "safe" positive samples, because a trajectory that only has a segment useful for the task of interest might not have the highest average $R'(s)$ due to its irrelevant part; however, an irrelevant trajectory will probably have the lowest $R'(s)$ throughout the whole trajectory, and thus can be identified as a "safe" negative sample. We collect such samples to form a new dataset $D_{\text{safe TA}}$.

Finally, the formal training of $c(s)$ uses states from $D_{\text{TS}}$ as positive samples and $D_{\text{safe TA}}$ as "safe" negative samples. The training objective of $c(s)$ is a combination of debiasing objective and standard cross entropy loss for binary classification, controlled by hyperparameter $\beta_2$. Specifically, we use

$$\min_{c(s)} \beta_2 L_2(c) + (1 - \beta_2) L_3(c), \text{ where}$$

$$L_2(c) = \min_{c(s)} -[\eta_p \mathbb{E}_{s \sim D_{\text{TS}}} \log c(s) + \max(0, \mathbb{E}_{s \sim D_{\text{safe TA}}} \log(1 - c(s)) - \eta_p \mathbb{E}_{s \sim D_{\text{TS}}} \log(1 - c(s)))],$$

$$L_3(c) = \mathbb{E}_{s \sim D_{\text{TS}}} \log c(s) + \mathbb{E}_{s \sim D_{\text{safe TA}}} \log(1 - c(s)).$$

(5)

In this work, we only consider $\beta_2 \in \{0, 1\}$; empirically, we found that $\beta_2 = 0$ is better if the agent's embodiments across $D_{\text{TS}}$ and $D_{\text{TA}}$ are the same, and $\beta_2 = 1$ is better if the embodiments differ. This is because a debiasing objective assumes positive samples still exist in the safe negatives, which pushes the classification margin further from samples in $D_{\text{TS}}$, and has a larger probability to classify expert segments in $D_{\text{TA}}$ as positive.

### 3.3 Trajectory-Aware Thresholding

With the discriminator $c(s)$ trained and $R(s) = \log \frac{c(s)}{1-c(s)}$ obtained, we can identify the most useful trajectory segments for our task. To do this, we use a simple thresholding which makes the weight employed in behavior cloning trajectory-aware. Formally, the weight $W(s_i, a_i)$ for weighted behavior cloning is calculated as

$$W(s_i, a_i) = \sum_{j=0}^{\infty} \gamma^j \exp(\alpha R(s_{i+j})),$$

(6)

where $(s_i, a_i)$ is the $i$-th step in a trajectory, and $\alpha$ is a hyperparameter that controls the strength of thresholding; $\alpha$ balances the tradeoff between excluding non-expert and including expert-trajectories. For an $i + j$ which exceeds the length of the trajectory, we set $s_{i+j}$ to be the last state of the (possibly incomplete) known trajectory, as the final state is of significant importance in many applications [18]. This design also allows us to conveniently address the example-based offline IL problem, where the final state is important. With the weights determined, we finally conduct a weighted behavior cloning with the objective $\max_\pi \mathbb{E}_{(s,a) \sim D_{\text{TA}}} W(s, a) \log \pi(a|s)$, where $\pi(a|s)$ is the desired policy.

## 4 Experiments

In this section, we evaluate TAILO on five different, challenging tasks across multiple mujoco testbeds. More specifically, we study the following two questions: 1) Is the algorithm indeed robust to incomplete trajectories in either task-agnostic (Sec. 4.1) or task-specific (Sec. 4.2) data, and does it work with little expert data in the task-agnostic dataset (Sec. 4.3)? 2) Can the algorithm also work well in example-based IL (Sec. 4.4) and learn from experts of different dynamics (Sec. 4.5)?

**Baselines.** We compare TAILO to four baselines: SMODICE [41], LobsDICE [23], ORIL [73], and Behavior Cloning (BC). Since LobsDICE works with state-pair occupancy, it cannot solve example-based IL; thus, we substitute LobsDICE in example-based IL with RCE [13], a state-of-the-art example-based RL method. Unless otherwise specified, we use 3 random seeds per method in each scenario.

**Environment Setup.** Following SMODICE [41], unless otherwise specified, we test our algorithm on four standard mujoco testbeds from the OpenAI Gym [5], which are the hopper, halfcheetah, ant, and walker2d environment. We use normalized average reward[1] as the main metric, where higher reward indicates better performance; for environments where the final reward is similar, fewer gradient steps in weighted behavior cloning indicates better performance. We report the change of the mean and standard deviation of reward with respect to the number of gradient steps.

**Experimental Setup.** For all environments, we use $\alpha = 1.25, \beta_1 = 0.8, \eta_p = 0.2$; $\beta_2 = 1$ if $D_{\text{TS}}$ and $D_{\text{TA}}$ are generated by different embodiments, and $\beta_2 = 0$ otherwise. We use $\gamma = 0.998$ unless otherwise specified. We use an exactly identical discriminator and policy network as SMODICE: for the discriminator $c(s)$ and $c'(s)$, we use a small Multi-Layer Perceptron (MLP) with two hidden layers, width 256, and tanh activation function. For actor $\pi$, we use an MLP with two hidden layers, width 256, and ReLU [2] activation function. For the training of $c(s)$ and $c'(s)$, we use a learning

---

[1]Normalization standard is according to D4RL [16], and identical to SMODICE [41].

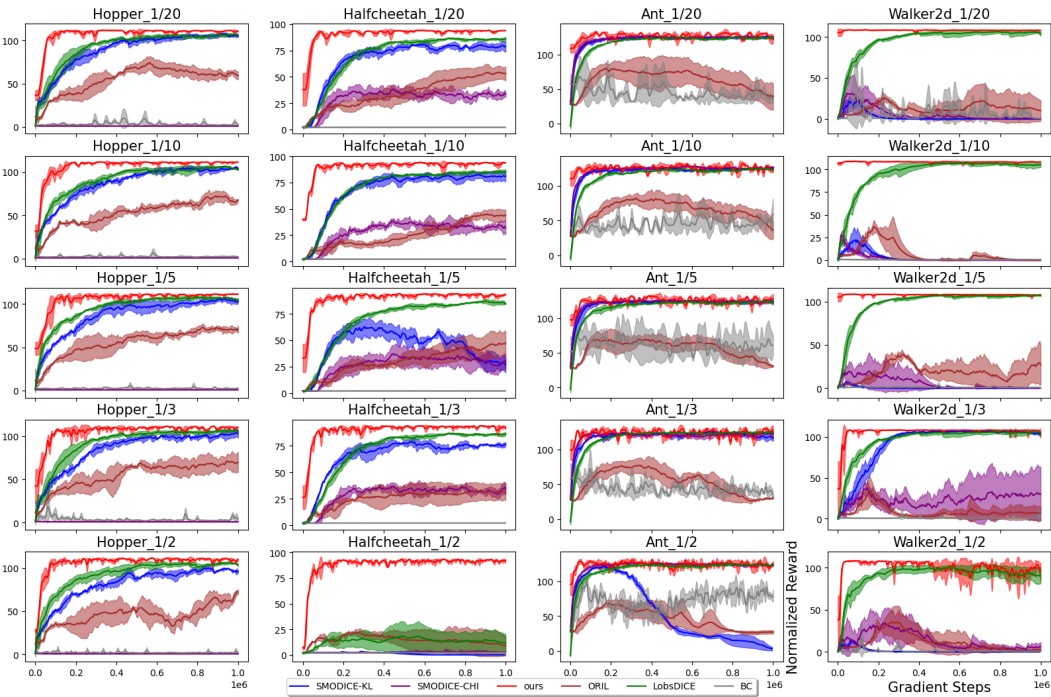

Figure 2: Reward curves for offline imitation learning with incomplete trajectories in the task-agnostic dataset, where the x-axis is the number of gradient steps and the y-axis is the normalized reward. The title for each sub-figure is in the format of "environment name"+"$1/x$" (task-agnostic data removed), where $x \in \{2, 3, 5, 10, 20\}$. We observe the proposed method to be the most stable.

rate of $0.0003$ and a 1-Lipschitz regularizer, run 10K gradient steps for $c'(s)$, and run 40K gradient steps for $c(s)$. For weighted BC, we use a learning rate of $10^{-4}$, a weight decay of $10^{-5}$, and run 1M gradient steps. For discriminator training, we use a batch size of $512$; for weighted BC steps, we use a batch size of $8192$. Adam optimizer [28] is used for both steps. See Appendix D for more details and Appendix F for a sensitivity analysis regarding the batch size, $\alpha, \beta_1, \beta_2, \eta_p$, and $\gamma$.

### 4.1 Learning from Task-Agnostic Dataset with Incomplete Trajectories

**Dataset Setup.** We modify the standard dataset settings from SMODICE to create our dataset. SMODICE uses offline datasets from D4RL [16], where a single trajectory from the "expert-v2" dataset is used as the task-specific data. The task-agnostic data consists of 200 expert trajectories (200K steps) from the "expert-v2" dataset and 1M steps from the "random-v2" dataset. Based on this, we iterate over the state-action pairs in the task-agnostic dataset, and remove one pair for every $x$ pairs. In this work, we test $x \in \{2, 3, 5, 10, 20\}$.

**Main Results.** Fig. 2 shows the result for different methods with incomplete task-agnostic trajectories, where our method outperforms all baselines and remains largely stable despite decrease of $x$ (i.e., increase of removed data), as the weights for each state-action pair do not change much. In the training process, we often witness SMODICE and LobsDICE to collapse due to diverging value functions (see Appendix C for explanation), which is expected; for runs that abort due to numerical error, we use a reward of $0$ for the rest of the gradient steps. Under a few cases, SMODICE with KL-divergence works decently well with larger noises (e.g., Halfcheetah_1/3 and Walker2d_1/3); this is because sometimes the smoothing effect of the neural network mitigates divergence. However, with larger batch size (See Fig. 23 in Appendix F.4.6) and more frequent and uniform updates on each data point, the larger the noise the harder SMODICE fails.

### 4.2 Learning from Task-Specific Dataset with Incomplete Trajectories

**Dataset Setup and Main Results.** We use SMODICE's task-agnostic dataset as described in Sec. 4.1. For the task-specific dataset, we only use the first $x$ steps and the last $y$

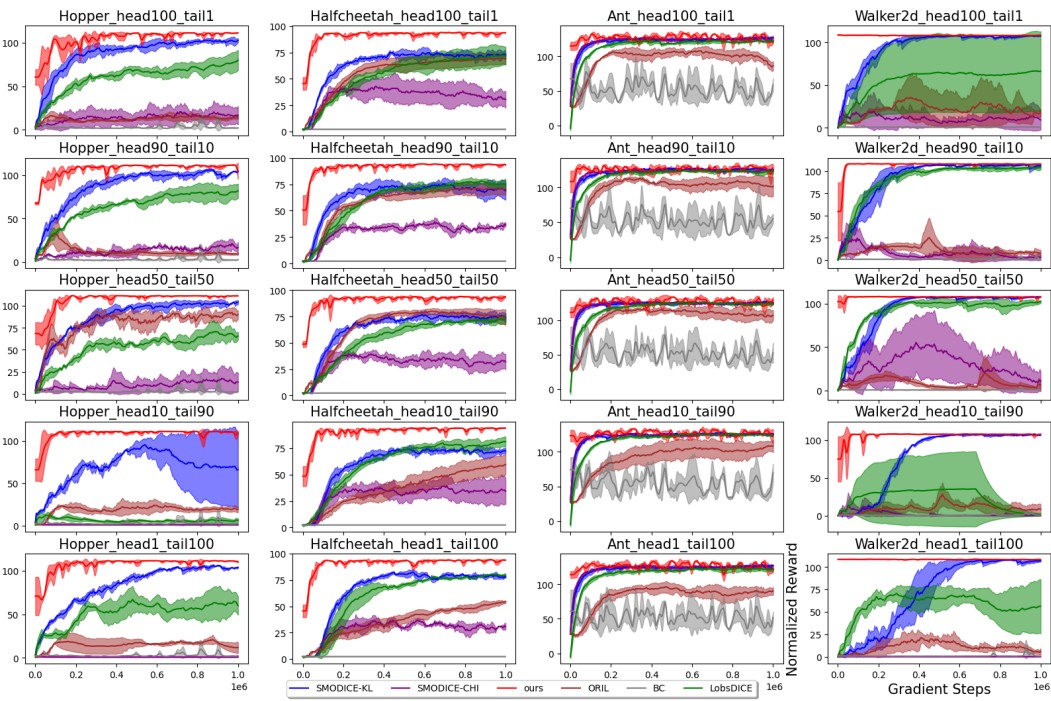

Figure 3: Reward curves for offline imitation with incomplete trajectories in the task-specific dataset. The title format for each subfigure is "environment name"+"head $x$"+"tail $y$" (states remained), where $(x, y) \in \{(100, 1), (90, 10), (50, 50), (10, 90), (1, 100)\}$. We observe the proposed method to be the most stable.

steps in the expert trajectory, and discard the remainder. In this work, we test $(x, y) \in \{(1, 100), (10, 90), (50, 50), (90, 10), (100, 1)\}$. Fig. 3 shows the result with incomplete task-specific trajectories, where our method outperforms all baselines and often achieves results similar to those obtained when using the entire task-specific dataset. In contrast, SMODICE and LobsDICE are expectedly unstable in this setting.

## 4.3 Standard Offline Imitation Learning from Observations

**Environment Setup.** In addition to the four standard mujoco environments specified above, we also test on two more challenging environments: the Franka kitchen environment and the antmaze environment from D4RL [16]. In the former, the agent needs to control a 9-DoF robot arm to complete a sequence of item manipulation subtasks, such as moving the kettle or opening the microwave; in the latter, the agent needs to control a robot ant to crawl through a U-shaped maze and get to a particular location. As the kitchen environment requires less steps to finish, we use $\gamma = 0.98$ instead of $0.998$.

**Dataset Setup.** As existing methods already solve the four mujoco environments with SMODICE's dataset settings in Sec. 4.1 quite well (see Appendix F for result), we test a more difficult setting to demonstrate TAILO's ability to work well with *few expert trajectories in the task-agnostic dataset*. More specifically, we use the same task-specific dataset, but only use $40$ instead of $200$ expert trajectories from the "expert-v2" dataset to mix with the 1M "random-v2" steps data and form the task-agnostic dataset. For the more challenging kitchen and antmaze environment, we use the identical dataset as SMODICE, where a single trajectory is used as the task-specific dataset. The task-agnostic dataset for the kitchen environment consists of expert trajectories completing different subtasks (both relevant and irrelevant) in different orders, and the task-agnostic dataset for antmaze consists of data with varied optimality. See Appendix D for details.

**Main Results.** Fig. 4 shows the result for different methods in standard offline imitation learning from observations. We find our method to outperform all baselines on hopper, halfcheetah, ant and walker2d. Results are comparable to the best baseline on kitchen and antmaze. In the experiment, we found ORIL to often diverge, and the performance of SMODICE varies greatly depending on the $f$-divergence: on hopper, halfcheetah and walker2d, KL-divergence is much better, while $\chi^2$-

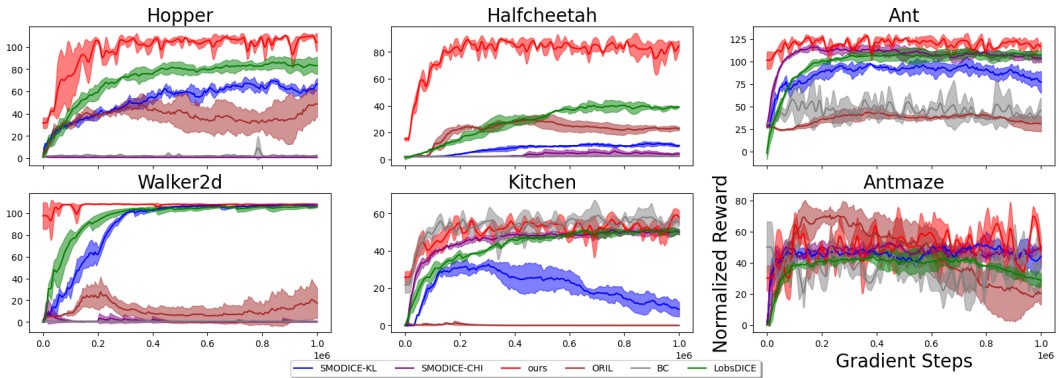

Figure 4: Reward curves for offline imitation learning from observations with few expert trajectories in the task-agnostic dataset. In all six environments, our method either outperforms or is comparable to the baselines.

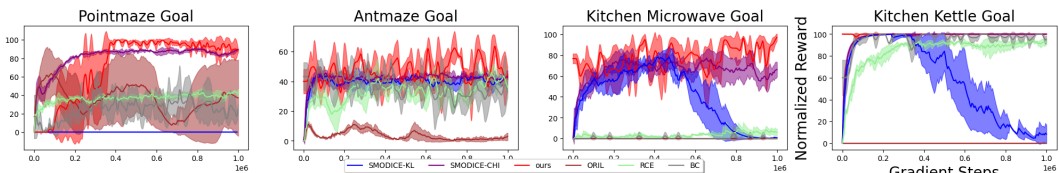

Figure 5: Reward curves for offline imitation learning from examples.

divergence is better on ant and kitchen. LobsDICE is marginally better than SMODICE, as the former considers state-pair occupancy instead of single state occupancy, which is more informative. Also worth noting: none of the methods exceeds a normalized reward of 60 in the kitchen environment; this is because the SMODICE experiment uses expert trajectories that are only expert for the first 2 out of all 4 subtasks. See Sec. F.3 in the Appendix for a detailed discussion.

## 4.4 Learning from Examples

**Environment Setup.** Following SMODICE, we test example-based offline imitation in three different testbeds: pointmaze, kitchen and antmaze. In the mujoco-based [59] pointmaze environment the agent needs to control a pointmass on a 2D plane to navigate to a particular direction. For the kitchen environment, we test two different settings where the agent is given successful examples of moving the kettle and opening the microwave respectively (denoted as "kitchen-kettle" and "kitchen-microwave"). As pointmaze requires less steps to finish, we use $\gamma = 0.98$ instead of $0.998$.

**Dataset Setup and Main Results.** Following SMODICE, we use a small set of success examples: $\leq 200$ states for antmaze and pointmaze and $500$ states for kitchen (see Appendix D for details) are given as the task-specific dataset. For pointmaze, the task-agnostic data contains 60K steps, generated by a script and distributed evenly along four directions (i.e., $25\%$ expert data); for other testbeds, the task-agnostic data is identical to SMODICE described in Sec. 4.1. Fig. 5 shows the results for offline imitation learning from examples on all environments tested in SMODICE; TAILO is marginally better than the baselines on pointmaze, antmaze, and kitchen-microwave, and is comparable (all close to perfect) on kitchen-kettle.

## 4.5 Learning from Mismatched Dynamics

**Environment and Dataset Setup.** Following SMODICE, we test our method on three environments: antmaze, halfcheetah, and ant. We use task-specific data from an expert with different dynamics (e.g., ant with a leg crippled; see Sec. E for details). Task-agnostic data follows SMODICE in Sec. 4.1.

**Main Results.** Fig. 6 shows the results for offline imitation learning from mismatched dynamics, where our method is the best in all three testbeds. Among the three environments, halfcheetah is the most difficult, as the state space for halfcheetah with shorter torso is unreachable by a normal

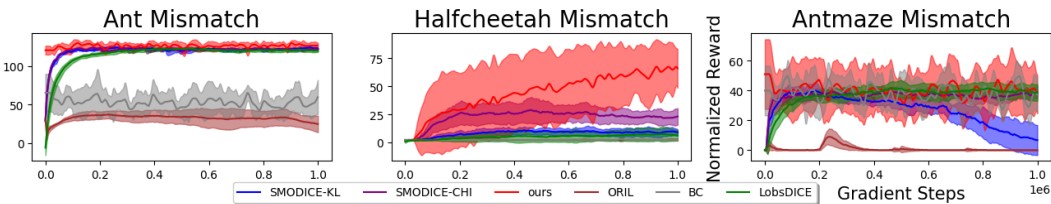

Figure 6: Reward curves for offline imitation learning from mismatched dynamics.

halfcheetah, i.e., the assumption that $d^{\text{TA}}(s) > 0$ wherever $d^{\text{TS}}(s) > 0$ in SMODICE and LobsDICE does not hold; in such a setting, our method is much more robust than DICE methods.

# 5   Related Work

**Offline Imitation Learning and DIstribution Correction Estimation (DICE).** Offline imitation learning aims to learn a policy only from data without interaction with the environment. This is useful where immature actions are costly, e.g., in a dangerous factory. The simplest solution for offline imitation learning is plain Behavior Cloning (BC) [51]. Many more methods have been proposed recently, such as BCO [60] and VMSR [33] (which pseudolabel actions), offline extensions of GAIL [3, 31, 61, 73], and similarity-based reward labeling [8, 35, 55, 64]. Currently, the state-of-the-art method for offline imitation learning is DIstribution Correction Estimation (DICE) [23, 27, 32, 34, 41], which minimizes the discrepancy between an expert's and a learner's state, state-action, or state-pair occupancy. Our method is inspired by DICE, but is much simpler and more effective. More recently, two works unify offline imitation learning with offline RL, which are offline-RL-based MAHALO [36] and DICE-based ReCOIL [56]. Different from SMODICE and LobsDICE, ReCOIL minimizes the divergence between learner and expert data mixed with non-expert data respectively, which removes the data coverage assumption. However, ReCOIL faces the problem discussed in Sec. 3.1 when dealing with incomplete trajectories, and MAHALO is based on state-pairs similar to LobsDICE, which cannot solve IL with incomplete trajectories or example-based IL like our TAILO.

**Learning from Observations and Examples.** Learning from observations (LfO) [60] requires the agent to learn from a task-specific dataset without expert action, which is useful when learning from videos [44] or experts with different embodiments [55], as the expert action is either unavailable or not applicable. Learning from examples is an extreme case of LfO where only the final goal is given [13]. There are three major directions: 1) pseudolabeling of actions which builds an inverse dynamic model and predicts the missing action [33, 60]; 2) occupancy divergence minimization with either DICE [23, 27, 34, 41, 71] or inverse-RL style iterative update [61, 67, 73]; and 3) RL/planning with reward assignment based on state similarity (often in visual imitation) [9, 55, 64]. Our proposed TAILO solves LfO with a simple solution different from existing ones.

**Discriminator as Reward.** The idea of training a discriminator to provide rewards for states is widely used in IL, including inverse RL methods [15, 22, 31, 73], DICE methods [23, 27, 41], and methods such as 2IWIL [65] and DWBC [68]. In this work, we propose a simple but explicit way to take the trajectory context into account, using the output from a discriminator as reward, which differs from prior works.

**Positive-Unlabeled Learning.** Positive-Unlabeled (PU) [11, 12, 29, 50] learning aims to solve binary classification tasks where only positive and unlabeled data are available. It is widely used in data retrieval [54], outlier detection [38], recommendation [7] and control tasks [67]. In this work, we utilize two PU learning achievements: the skill of identifying "safe" negative samples [39] and debiasing [11, 29]. The closest RL work to our use of PU learning is ORIL [73], which also uses positive-unlabeled learning to train a discriminator for reward estimation. However, there are three key differences between our method and ORIL: we define $R(s)$ differently for better thresholding, use different techniques for PU learning to prevent overfitting, and, importantly, the removal of value function learning. See Appendix F.6 for an ablation to demonstrate efficacy of these changes.

**Reward-Weighted Regression(RWR) [48] and Advantage-Weighted Regression(AWR) [45].** The idea in our Eq. (6) of weighted behavior cloning with weights being (often exponentiated) discounted return has been widely used in the RL community [1, 45, 63], and our objective resembles that of

RWR/AWR. However, our work differs from RWR/AWR inspired works [30, 43, 49] in the following aspects:

- The objective for both AWR and RWR are built upon the *related payoff procedure* [10, 21], which introduces an Expectation-Maximization (EM) procedure for RL. However, for offline IL in our case, iteration between E-step and M-step are infeasible. There are two workarounds for this: importance sampling and naively using one iteration. However, the former is known to be non-robust [42] and the latter, MARWIL [62], struggles in our testbeds (see Appendix F.2).

- We use neither an adaptive reward scaling term [48] in the EM framework nor parametric value estimation [45, 62], which are both widely utilized by RWR/AWR inspired works. However, the former is not guaranteed to preserve an optimal policy and thus is not an advantage [58], while the latter struggles in our setting where learning a good value function is hard, as illustrated by the performance of baselines such as DICE methods and MARWIL.

- For all existing RWR/AWR works, the reward labels are assumed to be available, which is different from our case.

## 6   Conclusion

We propose TAILO, a simple yet effective solution for offline imitation from observations by training a discriminator using PU learning, applying a score to each state-action pair, and then conducting a weighted behavior cloning with a discounted sum over thresholded scores along the future trajectory to obtain the weight. We found our method to improve upon state-of-the-art baselines, especially when the trajectories are incomplete or the expert trajectories in the task-agnostic data are few.

**Societal Impact.** Our work addresses sequential decision-making tasks from expert observations with fewer related data, which makes data-driven automation more applicable. This, however, could also lead to negative impacts on society by impacting jobs.

**Limitations and Future Directions.** Our method relies on the assumption that there exist expert trajectories or at least segments in the task-agnostic data. While this is reasonable for many applications, like all prior work with proprioceptive states, it is limited in generalizability. Thus, one promising future direction is to improve the ability to summarize abstract "skills" from data with better generalizability. Another future direction is learning from video demonstrations, which is a major real-world application for imitation learning from observations. Also, our experiments are based on simulated environments such as D4RL. Thus a gap between our work and real-life progress remains. While we are following the settings of many recent works, such as SMODICE [41], ReCOIL [56] and OTR [40], to bridge the gap using techniques such as sim2real in the robotics community [24] is another very important direction for future work.

**Acknowledgements.** This work was supported in part by NSF under Grants 2008387, 2045586, 2106825, MRI 1725729, NIFA award 2020-67021-32799, the Jump ARCHES endowment through the Health Care Engineering Systems Center, the National Center for Supercomputing Applications (NCSA) at the University of Illinois at Urbana-Champaign through the NCSA Fellows program, the IBM-Illinois Discovery Accelerator Institute, and the Amazon Research Award.

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

# Appendix: A Simple Solution for Offline Imitation from Observations and Examples with Possibly Incomplete Trajectories

The Appendix is organized as follows: first, we summarize our key findings. Then, in Sec. A, we provide a more rigorous introduction of the key mathematical concepts, and in Sec. B we provide the pseudocode for the training process of TAILO. In Sec. C, we explain why DICE methods struggle with incomplete trajectories in the task-agnostic or task-specific data. We then list additional implementation details of our method and the baselines in Sec. D, include additional experimental settings in Sec. E, and present additional experiment results as well as ablation studies in Sec. F; after that, we report the training time and computational resources utilized by each method in Sec. G. Finally, we examine the licenses of assets for our code in Sec. H. See https://github.com/KaiYan289/TAILO for our code.

The key findings of the additional experimental results are summarized as follows:

- **Can our method discriminate expert and non-expert trajectories in the task-agnostic dataset?** In Sec. F.1, we visualize the change of $R(s)$ and behavior cloning weight $W(s, a)$ along the trajectory, as well as the average $R(s)$ and $W(s, a)$ for expert and non-expert trajectories in the task-agnostic dataset $D_{\text{TA}}$. We find that our method successfully discriminates expert and non-expert trajectories in the task-agnostic dataset across multiple experiments.

- **How does our method perform compared to other offline RL/IL methods**? In addition to LobsDICE and SMODICE, in the appendix we compare our performance to the following offline RL/IL methods: a) a very recent DICE method, ReCOIL; b) methods with extra access to expert actions or reward labels, such as DWBC [68], model-based RL methods MOReL [26] and MOPO [70]; c) offline adaption for Advantage-Weighted Regression(AWR) [45], MARWIL [62]; d) a recent Wasserstein-based offline IL method, OTR [40]. We found our method to be consistently better than all these methods. See Sec. F.2 for details.

- **How does our method and baselines perform on the kitchen environment with "Kitchen-Complete-v0"?** The kitchen testbed adopted in SMODICE uses an expert trajectory from the "kitchen-complete-v0" environment in D4RL, which has a different subtask sequence from the ones evaluated in the environment and the ones in the task-agnostic data ("kitchen-mixed-v0"); the two environments also have different state spaces. When changed to "kitchen-complete-v0," all baselines except BC fail, because the state space between task-agnostic data and evaluation environment is different. Nonetheless our method still works. See Sec. F.3 for details.

- **How sensitive is our method to hyperparameters, such as $\alpha$, $\beta_1$, $\beta_2$, $\eta_p$, and $\gamma$?** We conduct a sensitivity analysis in Sec. F.4, and find our method to be generally robust to a reasonable selection of hyperparameters.

- **Our method and baselines use different batch sizes. Is this a fair comparison?** In Sec. F.4.6, we test our method with batch size 512 and SMODICE/LobsDICE with batch size 8192. We find that our method is more stable with a larger batch size for a few settings, but SMODICE and LobsDICE do not generally benefit from an increased batch size. In addition, our method with batch size 512 is still better than the baselines. Thus, the comparison is fair to SMODICE and LobsDICE.

- **How important is the Lipschitz regularizer for training of $R(s)$?** A Lipschitz regularizer is used in many methods tested in this work, including our method, SMODICE, LobsDICE, and ORIL. In Sec. F.4.7, we found this regularizer to be crucial for the generalization of the discriminator and $R(s)$.

- **Our standard imitation from observation setting is different from that of SMODICE. Is this fair?** Compared with the SMODICE's original setting, in Sec. 4.3 we focused on a more challenging experiment setting *with less expert trajectories* in the task-agnostic dataset for the four mujoco environments (hopper, halfcheetah, ant, and walker2d). In the appendix, we also provide experimental evaluation under the SMODICE's original setting, and find our method still outperforming baselines. See Sec. F.5 for details.

- **How is our method different from ORIL, and where does the performance gain come from?** Our method is different from ORIL in three major aspects: definition of $R(s)$, positive-unlabeled learning technique, and policy retrieval. In Sec. F.6, we report that each of the three aspects contributes to the reward increase; among them, policy retrieval is the most important factor, and the positive-unlabeled learning technique is the least.

## A  Mathematical Concepts

In this section, we rigorously introduce four mathematical concepts in our paper, which are state(-action/-pair) occupancy, debiasing objective for positive-unlabeled learning, $f$-divergences, and Fenchel conjugate. The first one is used throughout the paper, the second is used in the training of $c(s)$, and the others are used in Sec. C.

**State, State-Action, and State-pair Occupancy.** Consider an infinite horizon MDP $(S, A, T, r, \gamma)$ with initial state distribution $p_0$, where the state at the $t$-th timestep is $s_t$ and the action is $a_t$. Then, given any fixed policy $\pi$, the probability $\Pr(s_t = s)$ of landing in any state $s$ for any step $t$ is determined. Based on this, the state visitation frequency $d^\pi(s)$ with policy $\pi$, also known as *state occupancy*, is defined as $d^\pi(s) = \sum_{t=1}^\infty \Pr(s_t = s)$. Similarly, the state-action frequency is defined as $d^\pi(s, a) = \sum_{t=1}^\infty \Pr(s_t = s, a_t = a)$, and the state-pair frequency is defined as $d^\pi(s, s') = \sum_{t=1}^\infty \Pr(s_t = s, s_{t+1} = s')$. For better readability, with a little abuse of notation, we use the same $d$ for all three occupancies throughout the paper. In this work, we refer to the occupancy with the learner's policy $\pi$ as $d^\pi$, we refer to the occupancy with average policy of the task-agnostic dataset as $d^{\text{TA}}$, and to the occupancy with average policy of the task-specific dataset as $d^{\text{TS}}$.

**Debiasing Objective for Positive-Unlabeled Learning.** Consider a binary classification task with feature $x \in \mathbb{R}^n$ and label $y \in \{0, 1\}$. Assume we have access to both labeled positive dataset $D_P$ and labeled negative dataset $D_N$, and the *class prior*, i.e., the probability of having a label when uniformly sampled from the dataset, is $\eta_p = \Pr(y = 1)$ for positive labels and $\eta_n = \Pr(y = 0)$ for negative samples. Then, with sufficiently many samples, the most commonly used loss function is a cross entropy loss. The average cross entropy loss can be approximated by

$$\eta_p \mathbb{E}_{(x,y)\sim D_P}[-\log \hat{y}] + \eta_n \mathbb{E}_{(x,y)\sim D_N}[-\log(1 - \hat{y})], \tag{7}$$

where $\hat{y} = c(x)$ is the output of the discriminator.

In positive-unlabeled learning, we only have access to the positive dataset $D_P$ and unlabeled dataset $D_U$ as an unlabeled mixture of positive and negative samples. One naive approach is to regard all samples from $D_U$ as negative samples. While this sometimes works, it falsely uses the loss function for positive samples in $D_U$, and thus introduces *bias*. To avoid this and get better classification results, multiple debiasing objectives [11, 29] have been proposed. Such objectives assume that the unlabeled dataset $D_U$ consists of $\eta_p$ portion of positive data and $\eta_n$ portion of negative data. Thus, we have

$$\mathbb{E}_{(x,y)\sim D_U}[-\log(1 - \hat{y})] = \eta_p \mathbb{E}_{(x,y)\sim D_P}[-\log(1 - \hat{y})] + \eta_n \mathbb{E}_{(x,y)\sim D_N}[-\log(1 - \hat{y})]. \tag{8}$$

Correspondingly, Eq. (7) can be approximated by

$$-[\eta_p \mathbb{E}_{(x,y)\sim D_P} \log \hat{y} + \mathbb{E}_{(x,y)\sim D_U} \log(1 - \hat{y}) - \eta_p \mathbb{E}_{(x,y)\sim D_P} \log(1 - \hat{y})]. \tag{9}$$

This is the formulation utilized by ORIL [73]. However, as Kiryo et al. [29] pointed out, the red part in Eq. (9) is an approximation for $\eta_n \mathbb{E}_{(x,y)\sim D_N}[-\log(1 - \hat{y})]$, and thus should be no less than 0; violation of such a rule could lead to overfitting (see [29] for details). Therefore, we apply a max operator $\max(\cdot, 0)$ to the red part. The objective thus becomes

$$-[\eta_p \mathbb{E}_{(x,y)\sim D_P} \log \hat{y} + \max(\mathbb{E}_{(x,y)\sim D_U} \log(1 - \hat{y}) - \eta_p \mathbb{E}_{(x,y)\sim D_P} \log(1 - \hat{y}), 0)]. \tag{10}$$

Note that while Kiryo et al. [29] uses an alternative update conditioning on the red term, we found the max operator to be more effective in our case. For our first step of discriminator training, we use Eq. (10), with $D_{\text{TS}}$ as $D_P$, $D_{\text{TA}}$ as $D_U$, state $s$ as feature $x$, and action $a$ as label $y$. For the second step, we use Eq. (10) again for task-specific data with mismatch dynamics, but this time we use

$D_{\text{TA safe}}$ as $D_U$; otherwise, we use Eq. (7) with $D_{\text{TS}}$ as $D_P$, $D_{\text{TA safe}}$ as $D_N$, state $s$ as feature $x$, and action $a$ as label $y$.

**$f$-divergences.** The $f$-divergence is the basis of the DICE family of methods [23, 27, 32, 34, 41]. DICE methods minimize an $f$-divergence, such as the KL-divergence between the learner's policy and the average policy on the task-specific dataset. For any continuous and convex function $f$, any domain $\mathcal{X}$, and two probability distributions $p, q$ on $\mathcal{X}$, the $f$-divergence between $p$ and $q$ is defined as

$$D_f(p\|q) = \mathbb{E}_{x \sim q}[f(\frac{p(x)}{q(x)})]. \tag{11}$$

For example, for the KL-divergence, $f(x) = x \log x$, and $D_f(p\|q) = \text{KL}(p\|q) = \mathbb{E}_{x \sim q}[\frac{p(x)}{q(x)} \log \frac{p(x)}{q(x)}] = \mathbb{E}_{x \sim p} \log \frac{p(x)}{q(x)}$; for $\chi^2$-divergence, $f(x) = (x-1)^2$, and $D_f(p\|q) = \chi^2(p\|q) = \mathbb{E}_{x \sim q}(\frac{p(x)}{q(x)} - 1)^2 = \int_{x \sim \mathcal{X}} \frac{(p(x) - q(x))^2}{q(x)}$. Note that SMODICE uses $f(x) = \frac{1}{2}(x-1)^2$, which is essentially *half* $\chi^2$-divergence.

**Fenchel Conjugate.** For any vector space $\Omega$ with inner product $\langle \cdot, \cdot \rangle$ and convex and differentiable function $f : \Omega \to \mathbb{R}$, the Fenchel conjugate of $f(x), x \in \Omega$ is defined as

$$f_*(y) = \max_{x \in \Omega} \langle x, y \rangle - f(x). \tag{12}$$

In SMODICE, the derivation of the $\chi^2$-divergence uses $f(x) = \frac{1}{2}(x-1)^2$, and the Fenchel dual is $f_*(y) = \frac{1}{2}(y+1)^2$. However, such Fenchel dual is obtained with no constraint on $x$, while in SMODICE with KL-divergence, $x$ is constrained on a probability simplex. This extra relaxation could be a factor for its worse performance on some testbeds.

# B  Algorithm Details

Alg. 1 shows the pseudocode of our method.

# C  Why DICE Struggles with Incomplete Trajectories?

In our experiments, we empirically find that SMODICE and LobsDICE struggle with either incomplete task-specific or incomplete task-agnostic trajectories. We give an extended explanation in this section.

## C.1  Incomplete Task-Specific Trajectory

The phenomenon that DICE struggles with incomplete expert trajectories is first discussed in [72]: the work mentions that subsampled (i.e., incomplete) trajectories artifically "mask" some states and causes the failure of ValueDICE. Recent discussion indicates that such a failure on subsampled task-specific trajectories is closely related to overfitting [6] and a lack of generalizability [69, 72].

## C.2  Incomplete Task-Agnostic Trajectory

### C.2.1  KL-Based Formulation

The reason that SMODICE [41] with KL-divergence and LobsDICE [23] struggle with incomplete task-agnostic trajectories is more direct: consider the final objective of SMODICE with KL-divergence:

$$\min_V (1 - \gamma)\mathbb{E}_{s \sim p_0}[V(s)] + \log \mathbb{E}_{(s,a,s') \sim D_{\text{TA}}} \exp[R(s) + \gamma V(s') - V(s)], \tag{13}$$

where $p_0 \in \Delta(S)$ is the initial state distribution over the state space $S$, and "reward function" $R(s) = \log \frac{d^{\text{TS}}(s)}{d^{\text{TA}}(s)}$ is labeled by a discriminator $c(s)$ as described in Sec. 2.

Similarly, the final objective of LobsDICE is

$$\min_{V}(1-\gamma)\mathbb{E}_{s\sim p_0}V(s)+(1+\alpha)\log\mathbb{E}_{(s,a,s'))\sim D_{\text{TA}}}\exp[(\frac{1}{1+\alpha}(R(s,s')+\gamma V(s')-V(s))], \quad (14)$$

where the "reward function" $R(s,s')$ is based on a state pair instead of a single state, and $\alpha > 0$ is a hyperparameter. Note, both methods use a 1-sample estimate for the expectation of the dual variable $\mathbb{E}_{s'}V(s')$ for future state $s'$.

Assume that no two states in $D_{\text{TA}}$ are exactly the same, which is common in a high-dimensional continuous state space. In such a case, there are only two occurrences of $V(s)$ for a particular state $s \in D_{\text{TA}}$ in Eq. (13) and Eq. (14): for initial states, one in the linear term and the other in $-V(s)$ inside the exp-term; for other states, one in the $-V(s)$ term and the other in the $\gamma V(s')$ term inside the exp-term. Consider a trajectory $\tau = \{(s_1, a_1, s_2), (s_2, a_2, s_3), \ldots, (s_n, a_n, s_{n+1})\} \in D_{\text{TA}}$.[2] If $(s_2, a_2, s_3)$ is missing, we have the following ways to make up:

1. Use $D_{\text{TA}}$ without extra handling. In this case, the only occurrence of $V(s_3)$ will be $-V(s_3)$, as no future state exists for $s_2$. As the objective is *monotonic* (which is not the case in RL) with unconstrained $V(s_3)$, the convergence of the algorithm solely relies on smoothing from nearby states in $D_{\text{TA}}$. This is also the method that we tested in Sec. F.

2. Let the trajectory contain $(s_1, a_1, s_3)$. In this case, the method would learn with a wrong dynamic of the environment because $s_1$ cannot transit to $s_3$ by conducting action $a_1$ as such a ground truth would suggest;

3. Consider $s_2$ as a terminal state of the trajectory $\tau$, where the agent could learn to halt in the middle of an expert trajectory if $\tau$ is an expert trajectory;

4. Train an inverse dynamic model for pseudolabeling of the action. However, such a method cannot deal with more than one consecutive transition missing.

In conclusion, none of the workarounds described above addresses the concern. Note that the problem described here is also valid for ValueDICE [32] and many follow-up works such as DemoDICE [27] as long as the Donsker-Varadhan representation [4] of the KL-divergence is used. Fig. 7 illustrates this divergence, which empirically verifies our motivation.

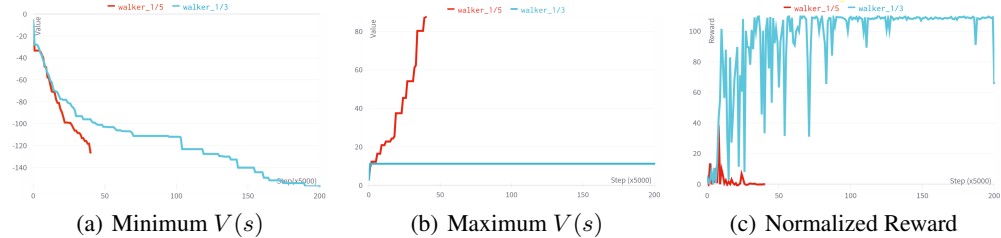

(a) Minimum $V(s)$        (b) Maximum $V(s)$        (c) Normalized Reward

Figure 7: Visualization of collapsing patterns of SMODICE-KL in Sec. 4.1 (walker_1/5 is collpasing; walker_1/3 is not). Walker_1/5 stops early due to NaN values in training, which we denote as 0 reward in our paper. It is clearly shown that the reward decrease is closely related to $V(s)$ divergence.

### C.2.2 $\chi^2$-Based Formulation

Empirically, we found that SMODICE with $\chi^2$-divergence struggles on several testbeds such as halfcheetah, hopper, and walker2d, which is consistent with the results reported by SMODICE. The performance gap could be due to the following two reasons:

First, violation of Theorem 1 in SMODICE [41]. Theorem 1 in SMODICE reads as follows:

**Theorem 1**. Given the assumption that $d^{\text{TA}}(s) > 0$ whenever $d^{\text{TS}}(s) > 0$, we have

---

[2]Because DICE methods utilize the next state $s'$ for a state-action pair $(s, a)$, we write the state-action trajectories as transition $(s, a, s')$ trajectories.

$$\text{KL}(d^\pi(s)\|d^{\text{TS}}(s)) \leq \mathbb{E}_{s \sim d^\pi}[\log(\frac{d^{\text{TA}}(s)}{d^{\text{TS}}(s)})] + \text{KL}(d^\pi(s,a)\|d^{\text{TA}}(s,a)), \tag{15}$$

and furthermore, for any $f$-divergence $D_f$ larger than KL,

$$\text{KL}(d^\pi(s)\|d^{\text{TS}}(s)) \leq \mathbb{E}_{s \sim d^\pi}[\log(\frac{d^{\text{TA}}(s)}{d^{\text{TS}}(s)})] + D_f(d^\pi(s,a)\|d^{\text{TA}}(s,a)). \tag{16}$$

The right hand side of Eq. (15) is the optimization objective of SMODICE with KL-divergence, and $\chi^2$-divergence is introduced via Eq. (16) as an upper bound.

However, SMODICE uses $f(x) = \frac{1}{2}(x-1)^2$ instead of $f(x) = (x-1)^2$ as the $\chi^2$-divergence; i.e., the $\chi^2$-divergence is *halved* in the objective. While the $\chi^2$-divergence is an upper bound of the KL-divergence [19], **half of the $\chi^2$-divergence is not**. For example, consider two binomial distributions $B(1,p)$ and $B(1,q)$. The KL-divergence between the two is $p \log \frac{p}{q} + (1-p) \log \frac{1-p}{1-q}$, and the $\chi^2$-divergence is $\frac{(p-q)^2}{q} + \frac{(q-p)^2}{1-q}$. When $p = 0.99$ and $q = 0.9$, the KL-divergence is $0.99 \log 1.1 + 0.01 \log 0.1 \approx 0.0713$, and the $\chi^2$-divergence is $\frac{0.09^2}{0.9} + \frac{0.09^2}{0.1} = 0.09$, where half of the $\chi^2$ divergence is smaller than the KL-divergence. Thus, SMODICE with $\chi^2$-divergence is not optimizing an upper bound of $\text{KL}(d^\pi(s)\|d^{\text{TS}}(s))$ and the performance is not guaranteed. In Fig. 8, we plot the reward curves of using halved and full $\chi^2$-divergence under the settings of Sec. F.5; we find that using full $\chi^2$-divergence significantly increases performance on the walker2d environment and performs similarly on other environments, though still much worse than our method.

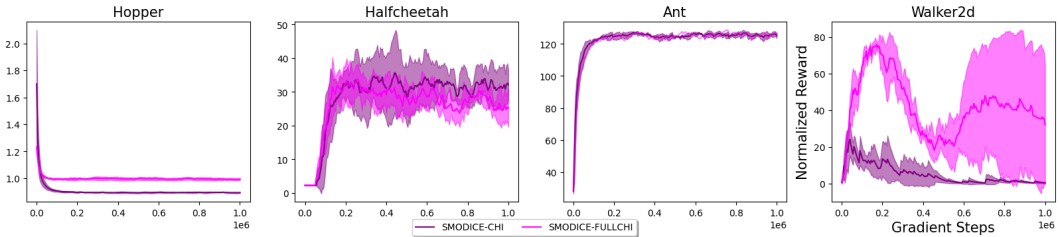

Figure 8: Performance comparison between using halved $\chi^2$-divergence (purple) adopted by SMODICE [41] and full $\chi^2$-divergence (fuchsia) with experiment settings in Sec. F.5. Full $\chi^2$-divergence performs significantly better on the walker2d environment than halved $\chi^2$-divergence, though still much worse than our method.

Second, SMODICE with $\chi^2$-divergence uses $f_*(y) = \frac{1}{2}(y+1)^2$ as the Fenchel conjugate of $f(x) = \frac{1}{2}(x-1)^2$. Such a conjugate is obtained when there is no constraint on $x$, i.e., $x \in \mathbb{R}$; however, such $x$, as indicated by the derivation of SMODICE with KL-divergence (Example 1 of Appendix C in SMODICE), should be on the probability simplex. This relaxation could be another reason for the performance drop in SMODICE with $\chi^2$-divergence.

### C.3 Other Recent Works

We noticed that there are also recent works in the DICE family that try to address the issue of incomplete (i.e., *subsampled*) trajectories, such as SparseDICE [6] and CFIL [14] for online imitation learning; LobsDICE [23] also discusses learning with subsampled expert trajectories. However, all those works only focus on incomplete task-specific (i.e. expert) trajectories instead of task-agnostic trajectories; also, our method can solve example-based IL and task-specific trajectories with all but one state provided at the beginning, which differs from their settings where the sampling of expert states/state-action pairs is uniform throughout the trajectory.

## D  Additional Implementation Details

Our code is provided in the supplementary material. We implement our algorithm from scratch, and use the implementation of SMODICE [41] (`https://github.com/JasonMa2016/SMODICE`) as

the codebase for SMODICE, ORIL [73], and RCE [13]. We obtain the code for LobsDICE [23] from their publicized supplementary material on OpenReview.

Tab. 1 summarizes the unique hyperparameters for our method, and Tab. 2 summarizes the common training paradigms for our method and the baselines. For all baselines, if the hyperparameter values in the paper and the code are different, we record the values from the code. We discuss the influence of batch size and sensitivity of our hyperparameters in Sec. E. Tab. 3 summarizes the hyperparameters specific to other methods. SMODICE, ORIL, and RCE first train a discriminator, and then jointly update actor and critic, while LobsDICE jointly updates all three networks.

# E Additional Experimental Settings

In this section, we describe in detail how the environment is setup and how the dataset is generated.

## E.1 Offline Imitation Learning from Task-Agnostic Dataset with Incomplete Trajectories

**Environment Settings.** Sec. 4.1 tests four mujoco environments: hopper, halfcheetah, ant, and walker2d. The detailed configuration for each environment is described as follows:

- **Hopper.** In this environment, an agent needs to control a 2D single-legged robot to jump forward by controlling the torques on its joints. The action space $A$ is $[-1, 1]^3$, one dimension for each joint; the state space $S$ is 11-dimensional, which describes its current angle and velocity. Note that the $x$-coordinate is not a part of the state, which means that the expert state approximately repeats itself periodically. Thus, in Fig. 10 the expert $R(s)$ is periodic. For this environment as well as halfcheetah, ant, and walker2d, the reward is gained by surviving and moving forward, and the episode lasts 1,000 steps.

- **Halfcheetah.** Similar to hopper, an agent needs to control a 2D cheetah-like robot with torques on its joints to move forward. The action space $A$ is $[-1, 1]^6$, and the state space $S$ is 17-dimensional describing its coordinate and velocity.

- **Ant.** In this environment, the agent controls a four-legged robotic ant to move forward in a 3D space with a 111-dimensional state space describing the coordinate and velocity of its joints, as well as contact forces on each joint. The action space is $[-1, 1]^8$.

- **Walker2d.** The agent in this environment controls a 2D two-legged robot to walk forward, with state space being 27-dimensional and action space being $[-1, 1]^8$.

Fig. 9 shows an illustration of each environment.

**Dataset Settings.** We generate our dataset on the basis of SMODICE. SMODICE uses 1 trajectory (1,000 states) from the "expert-v2" dataset in D4RL [16] as the task-specific dataset $D_{\text{TS}}$, and concatenates 200 trajectories (200K state-action pairs) from the "expert-v2" dataset and the whole "random-v2" dataset in D4RL (which contains $1M$ state-action pairs) to be the task-agnostic dataset $D_{\text{TA}}$.

Based on this, we use the task-specific dataset from SMODICE. For the task-agnostic dataset, we take the dataset from SMODICE, concatenate all state-action pairs from all trajectories into an array, and remove a state-action pair for every $x$ steps; we test $x \in \{2, 3, 5, 10, 20\}$ in this work.

## E.2 Offline Imitation Learning from Task-Specific Dataset with Incomplete Trajectories

**Environment Settings.** Sec. 4.2 tests hopper, halfcheetah, ant, and walker2d with the same environment settings as those discussed in Sec. E.1.

**Dataset Settings.** We use the task-agnostic dataset from SMODICE as described in Sec. E.1. For the task-specific dataset, we take the first $x$ and last $y$ steps from the task-specific dataset of SMODICE as the new task-specific dataset, and discard the other state-action pairs; in this work, we test $(x, y) \in \{(1, 100), (10, 90), (50, 50), (90, 10), (100, 1)\}$.

### E.3 Standard Offline Imitation Learning from Observation

**Environment Settings.** In addition to the four mujoco environments tested in Sec. E.1, Sec. 4.3 tests the kitchen and antmaze environments. The detailed configuration for the two environment is described as follows:

- **Kitchen.** In this environment, the agent controls a 9-DoF robotic arm to complete a sequence of 4 subtasks; possible subtasks include opening the microwave, moving the kettle, turning on the light, turning on the bottom burner, turning on the top burner, opening the left cabinet, and opening the right cabinet. The state space is 60-dimensional, which includes the configuration of the robot, goal location of the items to manipulate, and current position of the items.
- **Antmaze.** In this environment, the agent controls a robotic ant with 29-dimensional state space and 8-dimensional action space to move from one end of a u-shaped maze to the other end.

Fig. 9 illustrates the two environments.

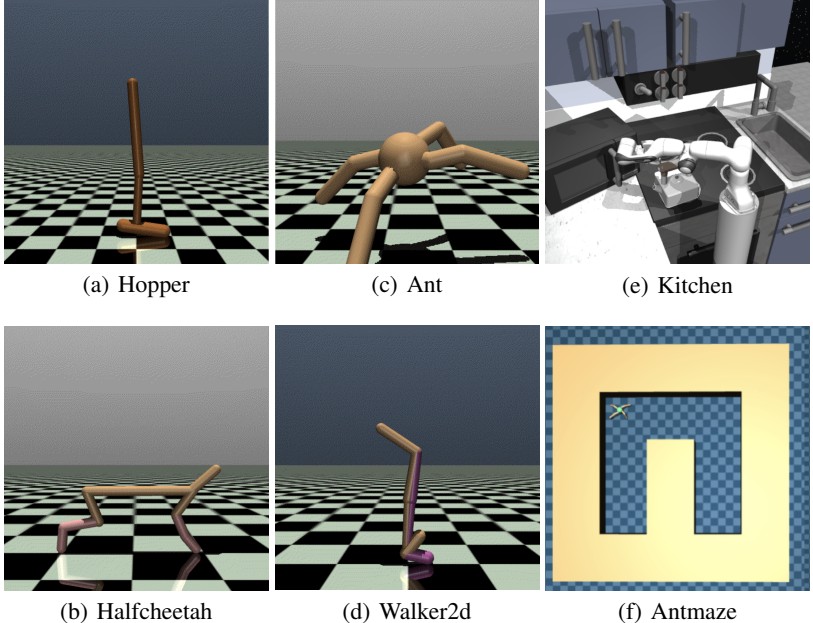

|  (a) Hopper | (c) Ant | (e) Kitchen |
| --- | --- | --- |
| (b) Halfcheetah | (d) Walker2d | (f) Antmaze |

Figure 9: Illustration of environments tested in Sec. 4.3 based on OpenAI Gym [5] and D4RL [16].

**Dataset Settings.** For the four mujoco environments, we take the task-specific dataset from SMODICE as described in Sec. E.1. As existing methods already perform well on those environments with SMODICE's dataset settings (see Sec. F.5 for results), we introduce a more challenging task-agnostic dataset concatenating 40 trajectories (40K state-action pairs) instead of 200 from the "expert-v2" dataset and the whole "random-v2" dataset in D4RL as the task-agnostic dataset $D_{\text{TA}}$.

For the more challenging kitchen and antmaze environments, we use identical dataset settings as SMODICE. For the kitchen environment, we use 1 trajectory (184 states) from the "kitchen-complete-v0" dataset as the task-specific dataset $D_{\text{TS}}$ and the whole "kitchen-mixed-v0" dataset as the task-agnostic dataset $D_{\text{TA}}$, which contains 33 different task sequences with a total of 136, 950 state-action pairs. For the antmaze environment, we use the expert trajectory of length 285 given by SMODICE as the task-specific dataset $D_{\text{TS}}$. We use the dataset collected by SMODICE as the task-agnostic dataset $D_{\text{TA}}$, which contains 1, 348, 687 state-action pairs.

### E.4 Offline Imitation Learning from Examples

**Environment Settings.** Sec. 4.4 tests antmaze and kitchen in the same environment settings as the ones discussed in Sec. E.3, but we only require the agent to complete one particular subtask instead

of all four (we test "opening the microwave" and "moving the kettle"). Additionally, Sec. 4.4 tests the pointmaze environment, which is a simple environment where the agent controls a pointmass to move from the center of an empty 2D plane to a particular direction. The action is 2-dimensional; the state is 4-dimensional, which describes the current coordinate and velocity of the pointmass.

**Dataset Settings.** For the antmaze environment, we use the same task-agnostic dataset as the one discussed in Sec. E.3, and we use 500 expert final states selected by SMODICE as the task-specific dataset. For the kitchen environment, we use the same task-agnostic dataset as the one discussed in Sec. E.3, and randomly select 500 states from the "kitchen-mixed-v0" dataset that has the subtask of interest completed as the task-specific dataset. For the pointmaze environment, we use a trajectory of 603 expert trajectories with 50K state-action pairs as the task-agnostic dataset, where the trajectories moving up, down, left and right each take up $1/4$. We use the last state of all trajectories that move left as the task-specific dataset.

### E.5 Offline Imitation Learning from Mismatched Dynamics

**Environment Settings.** Sec. 4.1 tests halfcheetah, ant, and antmaze with the same environment settings as the ones discussed in Sec. E.1 and Sec. E.3.

**Dataset Settings.** We use the task-agnostic dataset from Sec. E.1 (ant, halfcheetah) and Sec. E.3 (antmaze). For the task-specific dataset, we use the data generated by SMODICE, which is a single trajectory conducted by an agent with different dynamics. More specifically, for halfcheetah, we use an expert trajectory of length $1,000$ by a cheetah with a much shorter torso; for ant, we use an expert trajectory of length $1,000$ by an ant with one leg crippled; for antmaze, we use an expert trajectort of length 173 by a pointmass instead of an ant (and thus only the first two dimensions describing the current location are used for training the discriminator). See Appendix H of SMODICE [41] for an illustration of the environments.

## F  Additional Experiment Results

### F.1 Can Our Method Discriminate Expert and Non-Expert Data in the Task-Agnostic Dataset?

In Sec. 4, an important problem related to our motivation is left out due to page limit: Does the algorithm really succeed in discriminating expert and non-expert trajectories and segments in the task-agnostic data? We answer the question here with Fig. 10. It shows $R(s)$ and the weight for behavior cloning $W(s, a)$ along an expert and a non-expert trajectory in $D_{\text{TA}}$ on halfcheetah in Sec. 4.3, hopper with first 50 step and last 50 step as task-specific dataset in Sec. 4.2, and pointmaze in Sec. 4.4. We also plot the average $R(s)$ and weight $W(s, a)$ for state-action pairs in all expert trajectories and non-expert trajectories in $D_{\text{TA}}$. The result clearly shows that, even if the discriminator cannot tell the expert states from the non-expert states at the beginning of the episode because states from all trajectories are similar to each other, the weight successfully propagates from the later expert state to the early trajectory, and thus the agent knows where to go even when being close to the starting point; also, the weight $W(s, a)$ is smoother than the raw $R(s)$, which prevents the agent from being lost in the middle of the trajectory because of a few steps with very small weights.

### F.2 Comparison to Other Baselines

Besides the baselines tested in the main paper, we compare our method to a variety of other offline RL/IL methods. The tested methods include: a) a very recent method, ReCOIL; b) methods with extra access to expert actions or reward labels, such as DWBC [68], model-based RL methods MOReL [26] and MOPO [70]; c) offline adaption for Advantage-Weighted Regression (AWR) [45], MARWIL [62]; d) a recent Wasserstein-based offline IL method, OTR [40]. As we do not have the code for ReCOIL, and model-based RL methods takes a long time to run on our testbed, we test our method on their testbeds and use their reported numbers for comparison; for other methods, we test on our testbed (both under SMODICE setting in Sec. F.5 and standard LfO setting in Sec. 4.3). We found our method to consistently outperform all methods, even those with extra access to reward labels or expert actions.

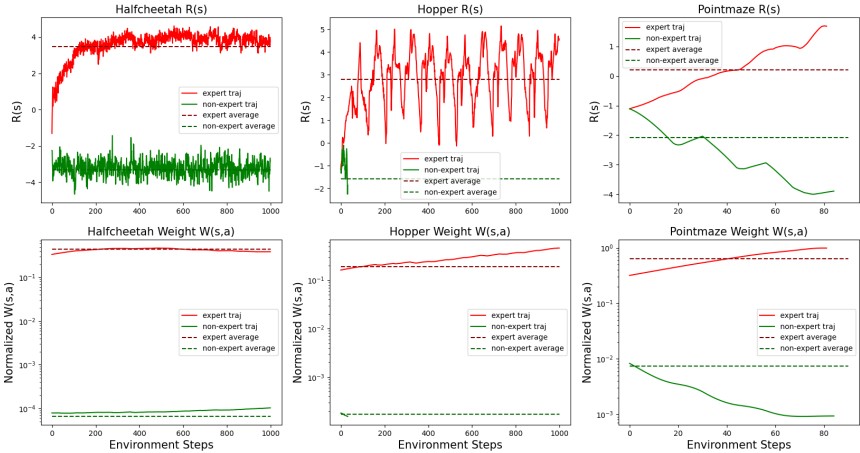

Figure 10: Illustration of $R(s)$, behavior cloning weight $W(s, a)$, and their average on expert and non-expert trajectories in $D_{\text{TA}}$. Curves for expert trajectories are plotted in red color, while non-expert trajectories are plotted in green color. The non-expert trajectory on the hopper environment is short because the agent topples quickly and the episode is terminated early. It is clear that our design of weights propagates the large weight from later expert states and smoothes the weight along the trajectory. Hopper has a large periodic change of $R(s)$, because the velocity vector for the joints of the agent is periodic.

### F.2.1 ReCOIL

Similar to Sec. 4.1, we test on the four common mujoco environments, which are hopper, halfcheetah, ant, and walker2d; the results we tested include random + expert (identical to SMODICE), random + few expert (30 expert trajectories instead of 40 in Sec. 4.3), and medium + expert (substitute the random dataset to medium dataset; see ReCOIL for details). The result of random + expert is illustrated in Fig. 26, where our method outperforms ReCOIL. The result of random + few expert is illustrated in Fig. 11, and the result of medium + expert is illustrated in Fig. 12. On all test beds, our method is significantly better than ReCOIL.

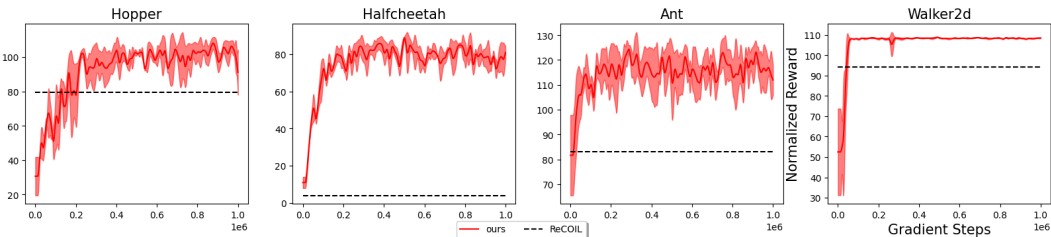

Figure 11: Performance comparison between our method and ReCOIL on the "random+few expert" testbed of ReCOIL.

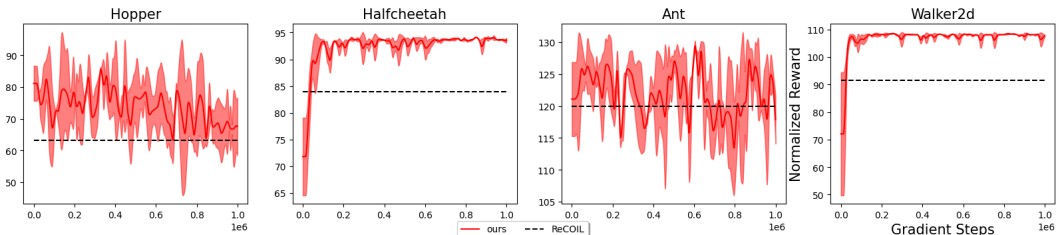

Figure 12: Performance comparison between our method and ReCOIL on the "medium+expert" testbed of ReCOIL.

### F.2.2 Model-Based RL

In this section, we compare our method to MOReL [26] and MOPO [70] which are provided with ground-truth reward labels, i.e., MOReL and MOPO have an advantage. The trajectory with the best return is provided as the task-specific dataset to our method. Tab. 4 shows the performance comparison between our method, MOReL and MOPO; despite being agnostic to reward labels, our method is still marginally better than MOReL (74.3 vs. 72.9 reward averaged over 9 widely tested environments {halfcheetah, hopper, walker2d} × {medium, medium-replay, medium-expert}), and much better than MOPO (74.3 vs. 42.1 average reward).

### F.2.3 DWBC, MARWIL and OTR

In this section, we additionally compare our method to three other baselines in offline IL: DWBC [68] that also trains a discriminator using Positive-Unlabeled (PU) learning, MARWIL [62] that is a naive adaptation of Reward-Weighted Regression [48] to offline scenarios with a similar actor objective as TAILO, and the recent Wasserstein-based method OTR [40] which computes Wasserstein distance between the task-specific and task-agnostic trajectories, assigns reward label based on optimization result and conducts offline RL. Among those methods, MARWIL and OTR can be directly applied to our scenario, while DWBC requires extra access to expert actions. Fig. 13 and Fig. 14 illustrate the result, respectively on the settings of Sec. F.5 and Sec. 4.3.

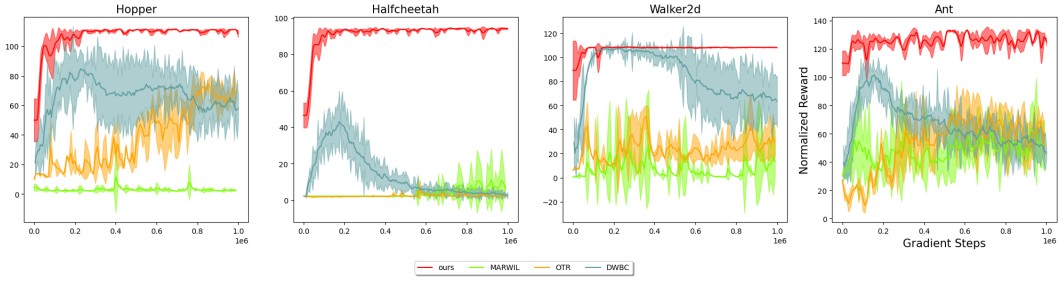

Figure 13: Performance comparison between our method and DWBC, MARWIL and OTR on settings of Sec. F.5. Our method outperforms all other methods, which all struggle in our settings.

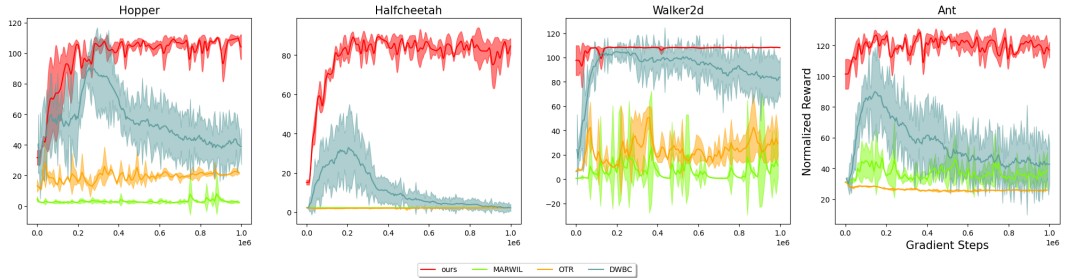

Figure 14: Performance comparison between our method and DWBC, MARWIL and OTR on settings of Sec. 4.3. Our method outperforms all other methods, which all struggle in our settings.

### F.3 Kitchen Environment Evaluated on "Kitchen-Complete-V0"

In Sec. 4.3, we mentioned that the SMODICE experiment uses expert trajectory that is only expert for the first 2 out of 4 subtasks. More specifically, SMODICE uses the expert trajectory in the "kitchen-complete-v0" environment in D4RL as the task-specific dataset, and uses data from the "kitchen-mixed-v0" environment as the task-agnostic dataset; they also evaluate on "kitchen-mixed-v0." However, the subtask list to complete in "kitchen-complete-v0" is {Microwave, Kettle, Light Switch, Slide Cabinet}, while the subtask list to complete in "kitchen-mixed-v0" is {Microwave, Kettle, Bottom Burner, Light Switch}.[3] Thus, the list of subtasks that is accomplished in expert

---

[3]See https://github.com/Farama-Foundation/D4RL/blob/master/d4rl/kitchen/__init__.py in D4RL and lines 29, 164 and 208 in https://github.com/JasonMa2016/SMODICE/blob/main/run_oil_observations.py for details.

trajectory (from "kitchen-complete-v0") and the list of subtasks evaluated by the environment (from "kitchen-mixed-v0") are only identical in the first two subtasks. We follow their setting in Sec. 4.3, and this is the reason why the normalized reward for any method is hard to reach 60 in Fig. 4, i.e., reach an average completion of $2.4$ tasks.

We now evaluate on "kitchen-complete-v0," and Fig. 15 shows the result when evaluating on this environment. The other settings remain identical to the ones discussed in Sec. 4.3. Note, this new evaluation setting introduces another challenge – since "kitchen-complete-v0" has a different set of subtasks to accomplish, the dimensions masked to zero differ; to be more specific, the 41st and 42nd dimensions are zero-masked in "kitchen-complete-v0" but not in "kitchen-mixed-v0," and vice versa for the 49th dimension. Thus, the task-agnostic data are in a different state space from the environment for evaluation. This significantly increases the difficulty of the environment, because the states met by the agent in evaluation are out of distribution from the task-agnostic data. Even under such a setting, our method as well as BC remains robust, while all the other baselines struggle.

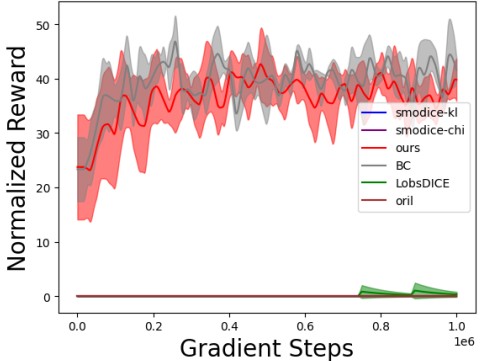

Figure 15: Reward curves for the kitchen environment evaluated on "kitchen-complete-v0"; only our method and BC are robust to the change of state space from task-agnostic data.

### F.4  Hyperparameter Sensitivity Analysis

In this section, we conduct a sensitivity analysis for the hyperparameters in our method.

#### F.4.1  Effect of $\alpha$

Fig. 16 illustrates the reward curves tested on the settings of Sec. 4.3 with different $\alpha$ (for scaling of $R(s)$), where the value we used throughout the paper is $\alpha = 1.25$. While an extreme selection of the hyperparameters leads to a significant decrease of performance (e.g., $\alpha = 2.0$ for antmaze), our method is generally robust to the selection of $\alpha$.

#### F.4.2  Effect of $\beta_1$

Fig. 17 shows the reward curves tested on the settings of Sec. 4.3 with different $\beta_1$ (ratio of "safe" negative samples), where the default setting is $\beta_1 = 0.8$. Our method is again generally robust to the selection of $\beta_1$. Obviously the reward decreases with an extreme selection of $\beta_1$, such as $\beta_1 = 0.3$. Intuitively, such robustness comes from two sources: 1) with small ratio of expert trajectories in the task-agnostic dataset ($< 5\%$), thus there is little expert data erroneously classified as safe negatives; 2) the Lipschitz-smoothed discriminator yields a good classification margin, and thus the $60\%+$ trajectories with lower average reward represent all non-expert data well.

#### F.4.3  Effect of $\beta_2$

In this work, we use $\beta_2 = 1$ when the embodiment of the task-specific dataset $D_{\text{TS}}$ is different from that of the task-agnostic dataset $D_{\text{TA}}$. We use $\beta_2 = 0$ otherwise. Ideally, after selecting safe negatives from the first step of the discriminator training, the obtained $D_{\text{TA safe}}$ should consist of (nearly) $100\%$ non-expert data, and thus we use $\beta_2 = 0$; however, when $D_{\text{TS}}$ is collected from a different embodiment, the recognition of "safe negatives" will be much harder, because the states reachable by the task-specific expert could be different from the agent that was used to collect task-agnostic

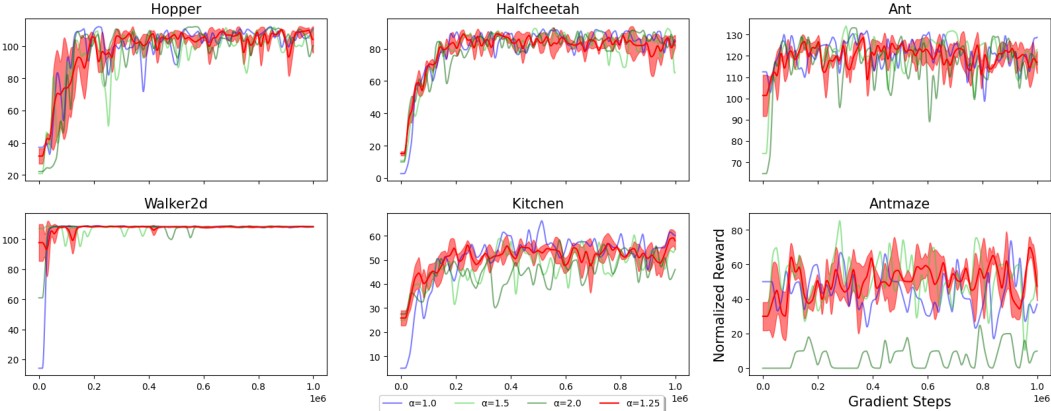

Figure 16: Reward curves tested in the settings of Sec. 4.3 with different $\alpha$. The default $\alpha = 1.25$ is plotted in red, lower $\alpha$ in blue, and higher $\alpha$ in green. The further $\alpha$ deviates from the default, the deeper the color. Our method is generally robust to the selection of $\alpha$.

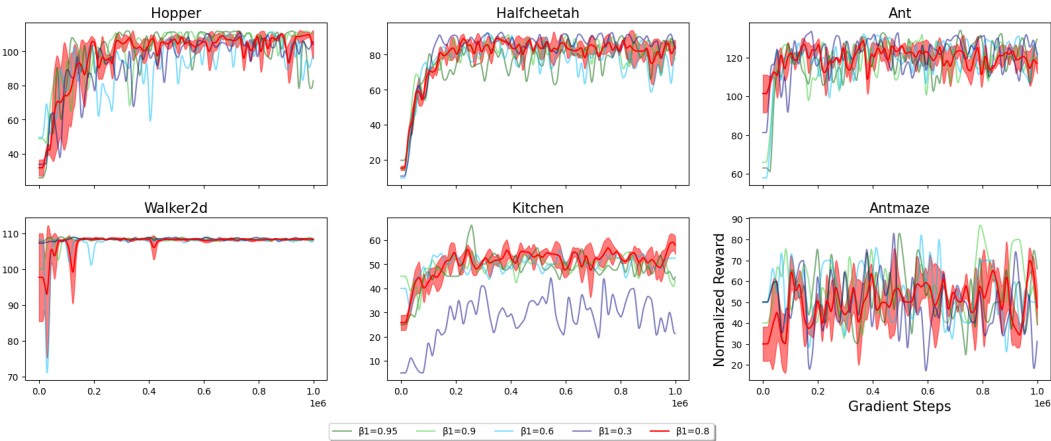

Figure 17: Reward curves tested in the settings of Sec. 4.3 with different $\beta_1$. The default $\beta_1$ is plotted in red, lower in blue and higher in green, with deeper color indicating further deviation from the default. Our method is generally robust to the selection of $\beta_1$.

data. Thus, we use the debiasing objective in the formal training step, i.e., $\beta_2 = 1$. We compare the reward of using $\beta_2 = 0$ and $\beta_2 = 1$ in Fig. 18, where $\beta_2 = 1$ works significantly better than $\beta_2 = 0$ on halfcheetah with mismatched dynamics.

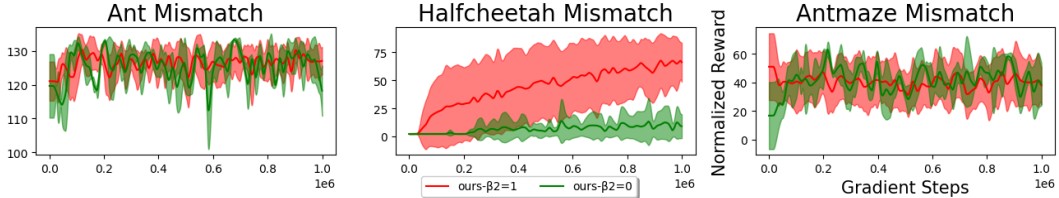

Figure 18: Reward curves for $\beta_2 = 1$ (i.e., debiasing objective) and $\beta_2 = 0$ (i.e., normal cross entropy loss) on environments with mismatching dynamics. $\beta_2 = 1$ is better than $\beta_2 = 0$ on the halfcheetah environment with mismatched expert dynamics.

### F.4.4 Effect of $\eta_p$

Fig. 19 shows the reward curves tested in the settings of Sec. 4.3 with different $\eta_p$, where the default setting is $\eta_p = 0.2$. The results show that our method is robust to the selection of $\eta_p$.

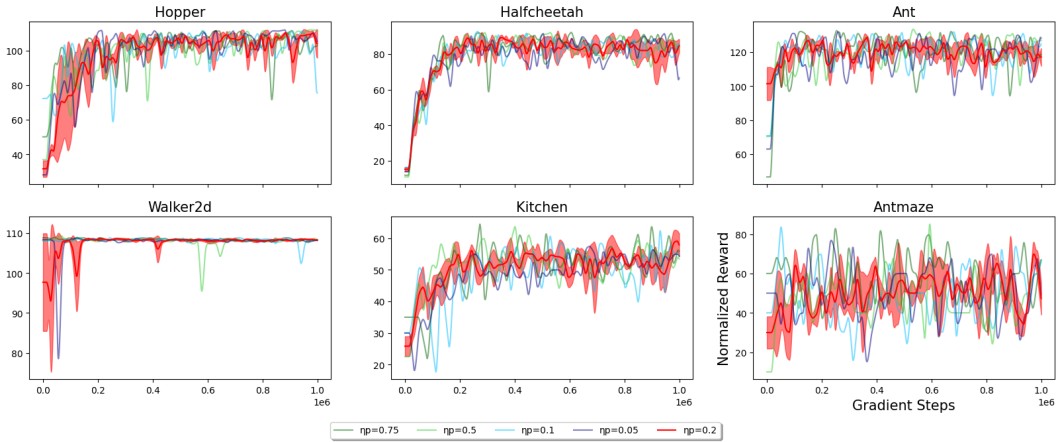

Figure 19: Reward curves tested on the settings of Sec. 4.3 with different $\eta_p$. The default $\eta_p$ is plotted in red, lower in blue and higher in green, with deeper color indicating further deviation from default. Our method is robust to the selection of $\eta_p$.

### F.4.5 Effect of $\gamma$

Fig. 20 shows the reward curves tested in the settings of Sec. 4.3 with different $\gamma$ (decaying factor for weight propagation along the trajectory); our default setting is $\gamma = 0.98$ for the kitchen and pointmaze environment, and $\gamma = 0.998$ otherwise. We found that when $\gamma$ is too low (e.g., 0.95), weights from the future expert state cannot be properly propagated to initial states where most trajectories are similar, and thus the algorithm is more likely to fail; more extremely, when $\gamma = 0$, our method works much worse as shown in Fig. 21 in both Sec. F.5 and Sec. 4.3 settings, which illustrates the necessity of propagation of future returns. Generally however, our method is robust to the selection of $\gamma$.

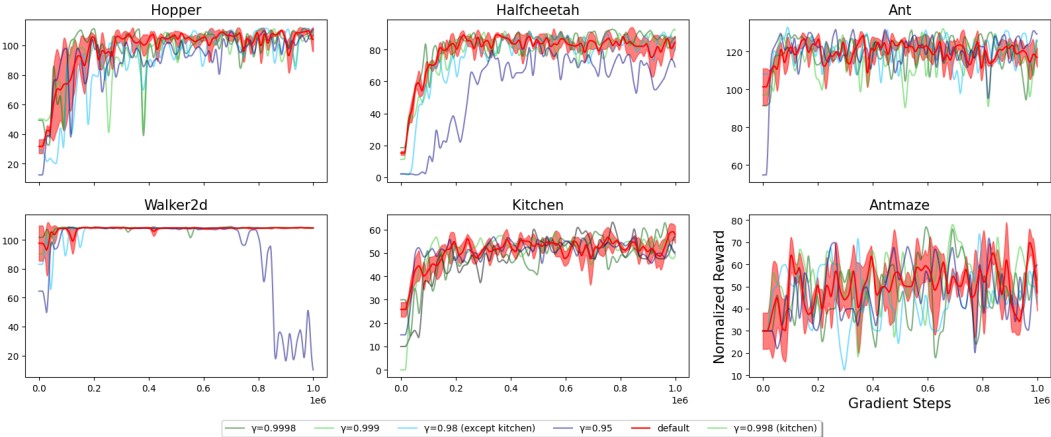

Figure 20: Reward curves tested in the settings of Sec. 4.3 with different $\gamma$. The default $\gamma$ is plotted in red, lower in blue and higher in green, with deeper color indicating further deviation from the default; note, kitchen uses a default $\gamma$ of 0.98 and other environments use 0.998. Our method is robust to the selection of $\gamma$.

### F.4.6 Effect of Batch Size

Since we use a different batch size (8192) than SMODICE and LobsDICE (512) for weighted behavior cloning, one possible concern is that the comparison might be unfair to the DICE methods, as we process more data. To address the concern, we test our algorithm with batch size 512 and SMODICE/LobsDICE with batch size 8192 in Sec. 4.1, Sec. 4.2, and Sec. 4.3. The results are illustrated in Fig. 22, Fig. 23, and Fig. 24. Generally, we found that on most testbeds, our method with a batch size of 512 is slightly less stable than the default batch size of 8192, but still on par or better than the baselines; meanwhile, SMODICE and LobsDICE with a batch size 8192 work

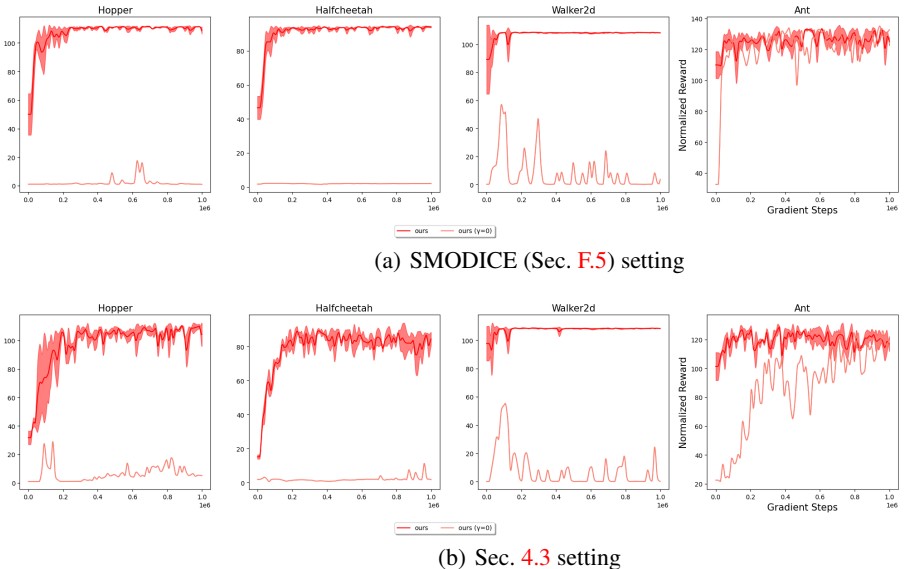

(a) SMODICE (Sec. F.5) setting

(b) Sec. 4.3 setting

Figure 21: Ablation of our method with $\gamma = 0$. $\gamma = 0$ struggles on both experiment settings of Sec. F.5 and Sec. 4.3, showing that $\gamma > 0$ is crucial to the performance of our method.

slightly better on Sec. 4.3 than batch size $512$, but are less stable in other scenarios such as Sec. 4.1 and Sec. 4.2. In conclusion, there is no general performance gain by increasing the batch size for SMODICE and LobsDICE. Thus, our comparison is fair to SMODICE and LobsDICE.

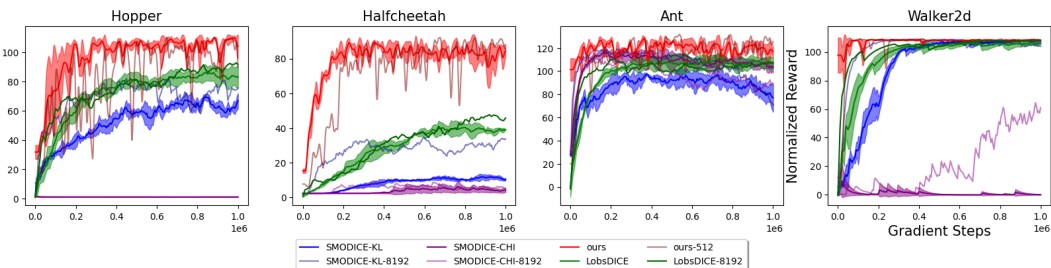

Figure 22: Performance comparison of SMODICE, LobsDICE, and our method with batch size $512$ and $8192$ on mujoco testbeds of standard offline imitation from observation (Sec. 4.3). Non-default settings (SMODICE/LobsDICE with batch size $8192$, ours with batch size $512$) are single lines without standard deviations, while default settings (SMODICE/LobsDICE with batch size $512$, ours with batch size $8192$) are with standard deviations. Our method with batch size $512$ works slightly worse, but is still better than baselines; baselines with batch size $8192$ work slightly better than use of a batch size of $512$.

### F.4.7 Effect of Lipschitz Regularizer

A Lipschitz regularizer on the discriminator is used for many methods tested in this paper, such as our method, SMODICE, LobsDICE, and ORIL. Intuitively, the regularizer makes the classification margin of the discriminator $c(s)$ smooth, and thus can give higher weights to expert trajectories in the task-agnostic dataset $D_{\text{TA}}$ instead of overfitting to the few states given in the task-specific dataset $D_{\text{TS}}$. In practice, we found this regularizer to be crucial; Fig. 25 shows the change of average $R(s) = \log \frac{c(s)}{1-c(s)}$ of all expert trajectories under the setting of Sec. 4.3 with respect to gradient steps. The result shows that $R(s)$ for the expert trajectories is extremely high when the Lipschitz regularizer is removed, which indicates severe overfitting and significantly worse performance.

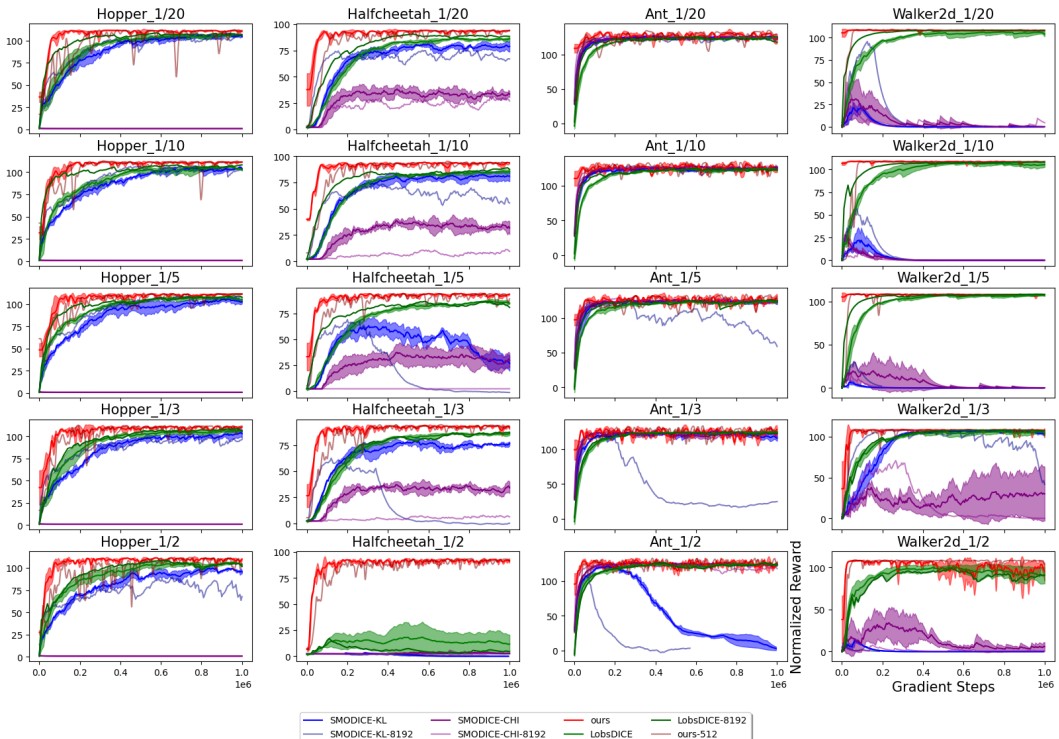

Figure 23: Performance comparison of SMODICE, LobsDICE and our method with batch size 512 and 8192 on mujoco testbeds of offline imitation with incomplete task-agnostic trajectories (Sec. 4.1). Note that SMODICE and LobsDICE with a larger batch size do not necessarily work better; in contrast, a larger batch size often leads to less stability on environments such as Hopper_1/2, Walker2d_1/2 and Ant_1/3. Our method with batch size 512 works slightly worse than 8192, but is still on par or better than the baselines.

## F.5 Performance Comparison Under Identical SMODICE Settings

In Sec. 4.3, we use less expert trajectory in $D_{\text{TA}}$ than the SMODICE setting. Fig. 26 shows the result where we use identical dataset settings as SMODICE, where our method is still the best among all baselines. Such a result is consistent with that reported in SMODICE. For ReCOIL [56], as the code is not available, we plot their average reward as a straight line; our method significantly outperforms ReCOIL.

## F.6 Ablation Comparing Our Method and ORIL

While our method and ORIL both utilize a discriminator-based $R(s)$ with positive-unlabeled learning, there are three major differences:

1. **Definition of** $R(s)$**.** For a discriminator $c(s)$ trained with non-expert samples as label $0$ and expert sample as label $1$, we use $R(s) = \log \frac{d^{\text{TS}}(s)}{d^{\text{TA}}(s)} = \log \frac{c(s)}{1-c(s)}$, while ORIL uses $R(s) = c(s)$.

2. **Positive-Unlabeled Learning Techniques.** The training of discriminator $C(s)$ in ORIL is only one step, and uses Eq. (9) as the training objective. In contrast, we first use Eq. (10) to find safe negative samples, and then use Eq. (10) or Eq. (7) (see Sec. A for details) to train the final discriminator $c(s)$ and consequently obtain $R(s)$.

3. **Policy Retrieval.** ORIL conducts RL on the offline dataset with reward labeled by $R(s)$, while our method uses a non-parametric approach to calculate coefficients for weighted behavior cloning.

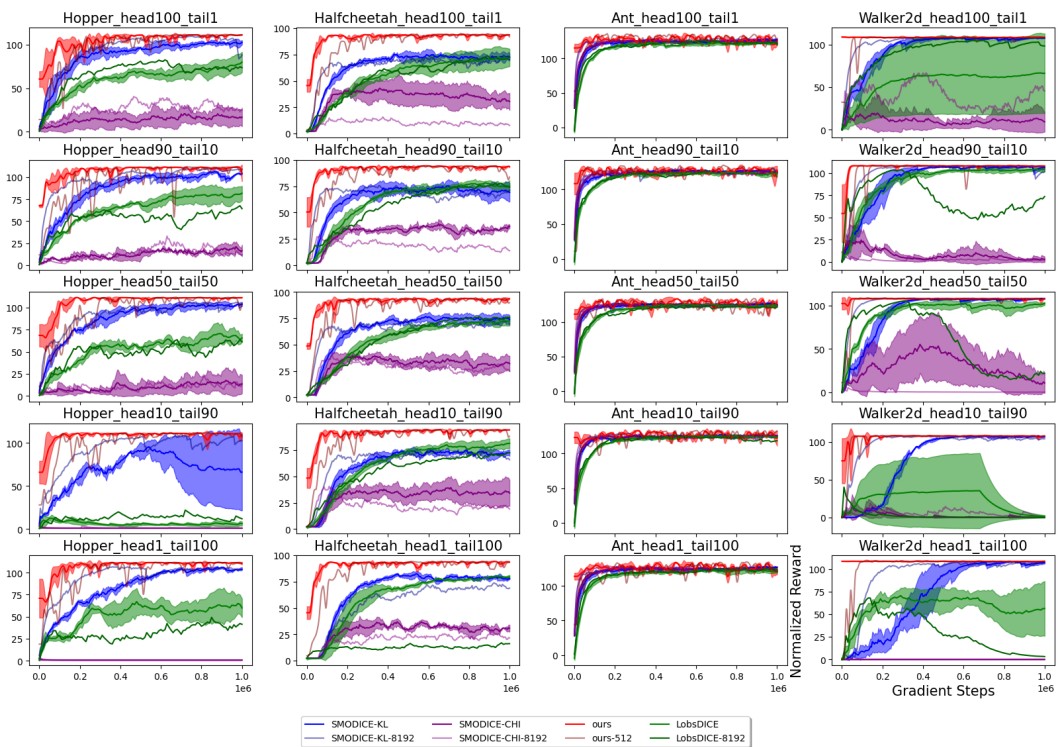

Figure 24: Performance comparison of SMODICE, LobsDICE and our method with batch size 512 and 8192 on mujoco testbeds of offline imitation with incomplete task-specific trajectories (Sec. 4.2). Again, larger batch size sometimes is worse for SMODICE and LobsDICE, e.g., for Halfcheetah_head1_tail100 and Hopper_head1_tail100. Our method with batch size 512 works slightly worse than 8192, but remains on par or better than the baselines.

We compare eight variants of the methods, denoted as ours, ours-V1, ours-V2, ours-V3, ORIL-logR-V1, ORIL-logR-V2, ORIL-01-V1, and ORIL-01-V2. Tab. 5 summarizes the differences between the eight variants.

The results are illustrated in Fig. 27. Generally, all variants except "ours" work at least marginally worse than "ours," which shows that all three differences matter in the final performance. Among the three variants of "ours," ours-v2 has the closest performance to "ours," which indicates that the positive-unlabeled learning technique is the least important factor, though it still makes a difference on environments such as halfcheetah with mismatched dynamics as shown in Fig. 28. Moreover, while ours-v1 (different $R(s)$) and ours-v3 (different $R(s)$ + ORIL positive-unlabeled learning technique) are significantly worse than "ours," ORIL-01-V1, ORIL-01-V2, ORIL-logR-V1, and ORIL-logR-V2 perform even worse and are only marginally different from each other. This indicates that retrieval of policy (RL vs. non-parametric) is the most important factor for the performance gap.

Additionally, in order to illustrate the importance of Positive-Unlabeled (PU) learning in general, we test the performance of our method without PU learning and find a significant performance decrease on some scenarios; Fig. 29 illustrates two of the failure cases without PU learning.

# G Computational Resource Usage

All our experiments are conducted on an Ubuntu 18.04 server with 72 Intel Xeon Gold 6254 CPUs @ 3.10GHz and a single NVIDIA RTX 2080Ti GPU. Given these resources, our method requires about $4 - 5$ hours to finish training in the standard offline imitation from observation scenario, while BC needs $3 - 4$ hours, ORIL, SMODICE, and LobsDICE need about $2.5 - 3$ hours. In our training process, training of $R(s)$ (both steps included) requires only 15-20 minutes. The inference speed for all methods is similar as the actor network is the same, and is thus not a bottleneck.

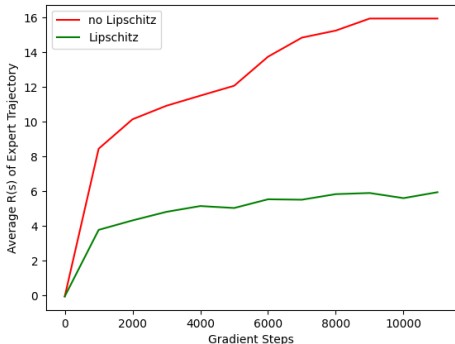

Figure 25: The average $R(s) = \log \frac{c(s)}{1-c(s)}$ of the expert trajectories in the task-agnostic dataset with respect to the gradient steps of discriminator training on the halfcheetah environment. We use the settings discussed in Sec. 4.3. The run without Lipschitz regularizer (red curve) overfits to the expert state, and thus has much higher average $R(s)$. Such a run diverges to infinity in later training.

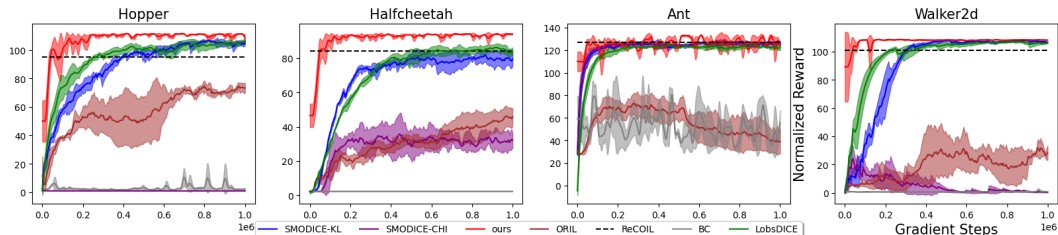

Figure 26: Reward curves for hopper, halfcheetah, ant, and walker2d evaluated under the identical settings as SMODICE; our method is the best among all methods including ReCOIL.

# H   Dataset and Algorithm Licenses

Our code is developed upon multiple algorithm repositories and environment testbeds.

**Algorithm Repositories.** We implement our method and behavior cloning from scratch. We test RCE, ORIL and SMODICE from the SMODICE repository, which has no license. We get LobsDICE code from their supplementary material of the publicized OpenReview submission, which also has no license.

**Environment Testbeds.** We utilize OpenAI gym [5], mujoco [59], and D4RL [16] as testbed, which have an MIT license, an Apache-2.0 license, and an Apache-2.0 license respectively.

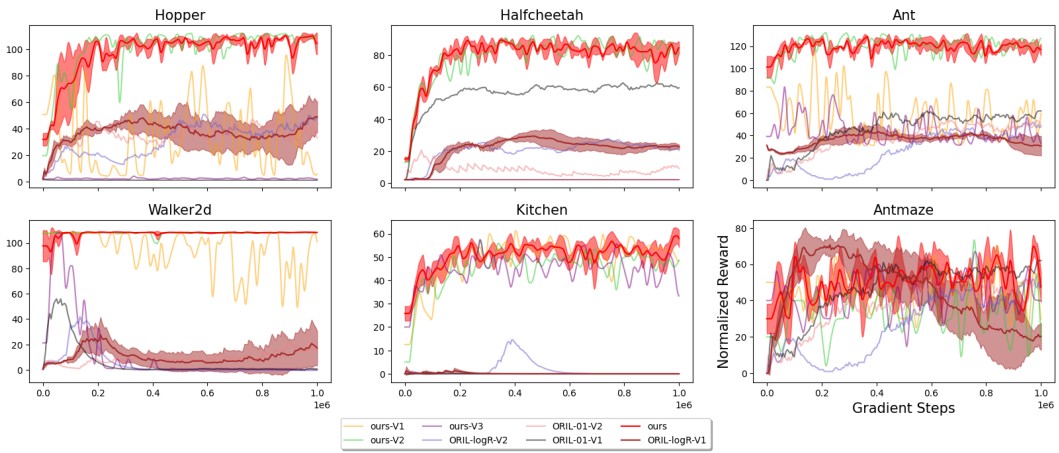

Figure 27: Ablation between our method and ORIL; see Tab. 5 for reference to the legend. While ours-V2 with ORIL-style positive-unlabeled learning has the closest performance to our final algorithm, it is still marginally worse. ORIL-01 and ORIL-logR fail, which shows that the retrieval of the policy is the most important factor for the performance gap.

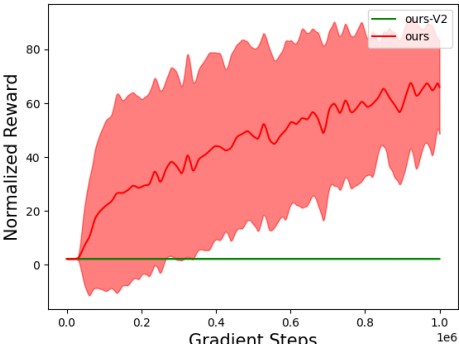

Figure 28: Comparison of ours and ours-V2 on the halfcheetah environment with mismatched dynamics (Sec. 4.5). Ours-V2 fails on this environment.

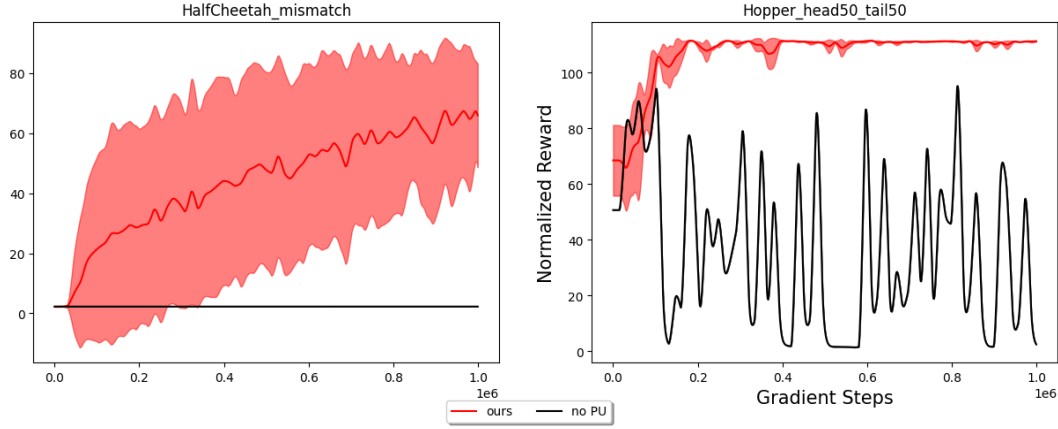

Figure 29: Examples of Failure cases of our method without PU learning; on the left is the halfcheetah environment with mismatched dynamics 4.5, and on the right is the hopper environment with only first and last 50 steps in the task-specific dataset (Sec. 4.2). In both cases, our method without PU learning struggles while our method with PU learning succeeds.

**Algorithm 1:** Our Algorithm, TAILO

---

**Input** : state-only task-specific dataset $D_{\text{TS}}$, state-action task-agnostic dataset $D_{\text{TA}}$ containing $m$ trajectories

**Input** : Discounted sum rate $\gamma$, ratio of "safe" negative sample $\beta_1$, positive prior $\eta_p$

**Input** : Number of gradient steps for pretraining $n_1$, formal training $n_2$, and number of epoches for weighted behavior cloning $n_3$

**Input** : Randomly initialized discriminator $c(s)$ and $c'(s)$, parameterized by $w$ and $w'$

**Input** : Learning rate $a$

**Output**: Learned policy $\pi_\theta(a|s)$, parameterized by $\theta$

**begin**

    // Pretraining Discriminator

1    **for** $i \in \{1, 2, \dots, n_1\}$ **do**

2        Sample $s_1 \sim D_{\text{TS}}, s_2 \sim D_{\text{TA}}$

3        $L_1 \leftarrow -[\eta_p \log c'(s_1) + \max(0, \log(1 - c'(s_2)) - \eta_p \log(1 - c'(s_1)))]$

4        $w' \leftarrow w' - a \cdot \frac{\partial L_1}{\partial w'}$

    // Labeling Safe Negative

5    **for** $i \in \{1, 2, \dots, m\}$ **do**            // Iterating through trajectories

6        **foreach** $s \in \tau_i$ **do**

            $R'(s) \leftarrow \log \frac{c'(s)}{1 - c'(s)}$

        $\bar{R}(\tau_i) \leftarrow \frac{\sum_{s \in \tau_i} R(s)}{|\tau_i|}$            // average over trajectory

7    $q \leftarrow \beta_1$ quantile of $\bar{R}$

8    $D_{\text{safeTA}} \leftarrow \{\tau | \bar{R}(\tau) < q\}$

    // Formal training of Discriminator

9    **for** $i \in \{1, 2, \dots, n_2\}$ **do**

10      Sample $s_1 \sim D_{\text{TS}}, s_2 \sim D_{\text{safeTA}}$

        **if** *Expert embodiment is different* **then**

11          $L \leftarrow -[\eta_p \log c(s_1) + \max(0, \log(1 - c'(s_2)) - \eta_p \log(1 - c'(s_1)))]$

        **else**

12          $L \leftarrow -[\log c(s_1) + \log(1 - c(s_2))]$

13      $w \leftarrow w - a \cdot \frac{\partial L}{\partial w}$

    // Assignment of Weights

14    **for** $i \in \{1, 2, \dots, m\}$ **do**         // Iterating through trajectories

15      **foreach** $s \in \tau_i$ **do**

16        $R(s) \leftarrow \log \frac{c(s)}{1 - c(s)}$

17        $W(s, a) \leftarrow \exp(\alpha R(s))$

18      $v \leftarrow \frac{R(s_{\text{goal}})}{1 - \gamma}$           // $s_{\text{goal}}$ is the last state in $\tau_i$

19      **foreach** $s \in \tau_i$ **do**         // This time in reverse

20        $W(s, a) \leftarrow v$

21        $v \leftarrow \gamma v + W(s, a)$

                                      // average over trajectory

    // Weighted Behavior Cloning

22    **for** $j \in \{1, 2, \dots, M_2\}$ **do**        // for loop over epochs

23      **foreach** $(s, a) \sim D_{TA}$ **do**      // for each data point

24        $L \leftarrow W(s, a) \pi_\theta(a|s)$

25        $\theta \leftarrow \theta - a \cdot \frac{\partial L}{\partial \theta}$

| Hyperparameter | Value | Meaning |
|---|---|---|
| $\alpha$ | 1.25 | Scaling factor for calculation of weights |
| $\beta_1$ | 0.8 | Estimated ratio of safe negative samples |
| $\beta_2$ | 1 (mismatch), 0 (others) | Whether to use debiasing objective in formal training |
| $\eta_P$ | 0.2 | Positive prior |
| $\gamma$ | 0.98 (kitchen, pointmaze), 0.998 (others) | decay factor in weight propagation along the trajectory |

Table 1: Hyperparameters specific to our method.

| Type | Hyperparameter | ours | BC | LobsDICE | SMODICE | ORIL | RCE |
|---|---|---|---|---|---|---|---|
| Disc. | Network Size | [256, 256] | N/A | [256, 256] | [256, 256] | [256, 256] | [256, 256] |
| | Activation Function | Tanh | N/A | ReLU | Tanh | Tanh | Tanh |
| | Learning Rate | 0.0003 | N/A | 0.0003 | 0.0003 | 0.0003 | 0.0003 |
| | Weight Decay | 0 | N/A | 0 | 0 | 0 | 0 |
| | Training Length | (10+40)K steps | N/A | 1M steps | 1K steps | 1K steps | 1K steps |
| | Batch Size | 512 | N/A | 512 | 256 | 256 | 256 |
| | Optimizer | Adam | N/A | Adam | Adam | Adam | Adam |
| Actor | Network Size | [256, 256] | [256, 256] | [256, 256] | [256, 256] | [256, 256] | [256, 256] |
| | Activation Function | ReLU | ReLU | ReLU | ReLU | ReLU | ReLU |
| | Learning Rate | 0.0001 | 0.0001 | 0.0003 | 0.0003 | 0.0003 | 0.0003 |
| | Weight Decay | $10^{-5}$ | $10^{-5}$ | 0 | 0 | 0 | 0 |
| | Training length | 1M steps | 1M steps | 1M steps | 1M steps | 1M steps | 1M steps |
| | Batch Size | 8192 | 8192 | 512 | 512 | 512 | 512 |
| | Optimizer | Adam | Adam | Adam | Adam | Adam | Adam |
| | Tanh-Squashed | Yes | Yes | Yes | Yes | Yes | Yes |
| Critic | Network Size | N/A | N/A | [256, 256] | [256, 256] | [256, 256] | [256, 256] |
| | Activation Function | N/A | N/A | ReLU | ReLU | ReLU | ReLU |
| | Learning Rate | N/A | N/A | 0.0003 | 0.0003 | 0.0003 | 0.0003 |
| | Weight Decay | N/A | N/A | 0.0001 | 0.0001 | 0 | 0 |
| | Training Length | N/A | N/A | 1M steps | 1M steps | 1M steps | 1M steps |
| | Batch Size | N/A | N/A | 512 | 512 | 512 | 512 |
| | Optimizer | N/A | N/A | Adam | Adam | Adam | Adam |
| | Discount Factor | N/A | N/A | 0.99 | 0.99 | 0.99 | 0.99 |

Table 2: Training paradigms for our method and baselines; Disc. is the abbreviation for discriminator. $[256, 256]$ in network size means a network with two hidden layers and width 256. For our method, $(10 + 40)$K means 10K gradient steps for the first step and 40K for the second step. Tanh-squashed means a Tanh applied at the end of the output to ensure the output action is legal.

| Method | Hyperparameter | Value | Notation |
|---|---|---|---|
| LobsDICE | Regularization factor | 0.1 | $\alpha$ |
| RCE, ORIL | RL algorithm | TD3 [17] | |
| | Policy Update Frequency | 2 | |
| | Policy Noise | 0.2 | |
| | Noise Clip | $[-0.5, 0.5]$ | |
| | Target Network Update Rate | 0.005 | $\tau$ |

Table 3: Unique hyperparameters for other methods.

| Environment | MOReL | MOPO | TAILO (Ours) |
|---|---|---|---|
| Halfcheetah-Medium | 42.1 | **42.3** | 39.8 |
| Hopper-Medium | **95.4** | 28 | 56.2 |
| Walker2d-Medium | **77.8** | 17.8 | 71.7 |
| Halfcheetah-Medium-Replay | 40.2 | **53.1** | 42.8 |
| Hopper-Medium-Replay | **93.6** | 67.5 | 83.4 |
| Walker2d-Medium-Replay | 49.8 | 39.0 | **61.2** |
| Halfcheetah-Medium-Expert | 53.3 | 63.3 | **94.3** |
| Hopper-Medium-Expert | 108.7 | 23.7 | **111.5** |
| Walker2d-Medium-Expert | 95.6 | 44.6 | **108.2** |
| Average | 72.9 | 42.1 | **74.3** |

Table 4: The performance comparison between our method and model-based RL methods on D4RL mujoco offline datasets. Our method works even better than the offline RL methods with extra access to the underlying reward label.

| Name | $R(s)$ | PU learning | Policy Retrieval |
|---|---|---|---|
| ours | $\log \frac{c(s)}{1-c(s)}$ | two-step with $\max(\cdot, 0)$ | non-parametric |
| ours-V1 | $10c(s)$ | two-step with $\max(\cdot, 0)$ | non-parametric |
| ours-V2 | $\log \frac{c(s)}{1-c(s)}$ | one step without $\max(\cdot, 0)$ | non-parametric |
| ours-V3 | $10c(s)$ | one step without $\max(\cdot, 0)$ | non-parametric |
| ORIL-01-V1 | $c(s)$ | one step without $\max(\cdot, 0)$ | RL |
| ORIL-logR-V1 | $\log \frac{c(s)}{1-c(s)}$ | one step without $\max(\cdot, 0)$ | RL |
| ORIL-01-V2 | $c(s)$ | two-step with $\max(\cdot, 0)$ | RL |
| ORIL-logR-V2 | $\log \frac{c(s)}{1-c(s)}$ | two-step with $\max(\cdot, 0)$ | RL |

Table 5: Configurations of the variants ablated in Fig. 27. Ours-V1 and ours-V3 use $10c(s)$ instead of $c(s)$ to scale the behavior cloning weight so as to be similar to "ours."

