# OpenReview forum: "A Simple Solution for Offline Imitation from Observations and Examples with Possibly Incomplete Trajectories"
_NeurIPS.cc/2023/Conference — NeurIPS 2023 poster_

### Official Review · Reviewer_Re2g · 2023-07-02

**Soundness:** 2 fair
**Presentation:** 2 fair
**Contribution:** 2 fair
**Rating:** 4
**Confidence:** 4

**Summary:**

This paper introduces Trajectory-Aware Imitation Learning from Observations (TAILO) for offline imitation learning. TAILO tackles the problem of learning from incomplete trajectories, where other state-of-the-art (SOTA) methods fail. Specifically, TAILO proposes a simple yet effective solution. It first learns to recover the task reward function by distinguishing the expert demonstrations and the sub-optimal data. In addition, TAILO treats the non-expert data as an unlabeled dataset, and relabelles the reward of sub-optimal transitions to provide more training data for the policy. Next, TAILO performs policy learning by weighted behavior cloning with accumulative return along a successful trajectory segment. In the experiments, TAILO successfully outperforms a series of baselines on offline imitation learning with incomplete trajectories.

**Strengths:**

- The paper studies an interesting problem with potential real-world applications.
- The empirical performance looks promising.


**Weaknesses:**

There are several main weaknesses of the paper.

Firstly, the presentation of the paper still has room for improvement. Specifically, I would suggest the authors double check the literature and be careful when making statements. There are several important but unfortunately erroneous statements in the paper. For example, the very first sentence in the abstract:

> Offline imitation from observations aims to solve MDPs where only task-specific expert states and task-agnostic non-expert state-action pairs are available.

This statement is inaccurate as by its definition, offline imitation learning discusses the case where a policy is learned with only off-policy data, excluding the on-policy environment interactions. Consider the standard behavior cloning, for example. The use of non-expert transitions is just one of the solutions to assist policy learning, as in ValueDICE [1], DemoDICE [2], and etc.
Also, in line 113, the authors state that there are three main steps for DICE methods:

> DICE methods consist of three parts: reward generation, optimization of the 114 value function V(s), and weighted behavior cloning.

This is inaccurate as well. For example, ValueDICE [1] directly learns the value function, rather than the rewards.
Besides the presentation issues, the contribution of the paper is relatively minor. The use of cumulative return / advantage function is standard in reinforcement learning literature. Mathematically, only by maximizing the cumulative return would give a correct policy improvement step. If we consider the specific form of weighted behavior cloning, or more precisely, advantage weighted regression, we can still find many existing works with similar structure [3, 4, 5].

There are some potential issues with the mathematical correctness and experiment details, which I will reserve for the questions.

References:

[1] Kostrikov, I., Nachum, O., & Tompson, J. (2020). Imitation learning via off-policy distribution matching. In International Conference on Learning Representations.

[2] Kim, G. H., Seo, S., Lee, J., Jeon, W., Hwang, H., Yang, H., & Kim, K. E. (2022, October). Demodice: Offline imitation learning with supplementary imperfect demonstrations. In International Conference on Learning Representations.

[3] Abdolmaleki, A., Springenberg, J. T., Tassa, Y., Munos, R., Heess, N., & Riedmiller, M. (2018). Maximum a posteriori policy optimisation. arXiv preprint arXiv:1806.06920.

[4] Peng, X. B., Kumar, A., Zhang, G., & Levine, S. (2019). Advantage-weighted regression: Simple and scalable off-policy reinforcement learning. arXiv preprint arXiv:1910.00177.

[5] Wang, Z., Novikov, A., Zolna, K., Merel, J. S., Springenberg, J. T., Reed, S. E., ... & de Freitas, N. (2020). Critic regularized regression. Advances in Neural Information Processing Systems, 33, 7768-7778.


**Questions:**

- In Sect. 3.2, the authors propose to learn two discriminators to label the non-expert trajectories and learn the reward function in an inverse reinforcement learning manner. However, it is common that solving a min-max game is naturally unstable and hard to optimize. Coupling two min-max objectives would even make it worse in my opinion. Are there any tricks to improve the stability? What are the failure cases of such an optimization scheme?
- In Eqn. 6, the authors propose to weight the log-likelihood by $\sum\limits_{j=0} \gamma^j \exp(\alpha R(s_{i+j}))$. This seems a bit odd to me. If we treat $R(s)$ as the exact reward function and solve the policy improvement by RL as inference, we will end up with exactly the AWR [1] objective, which is $\exp(Q(s, a) - V(s)) = \exp(\sum\limits_{j=0}\gamma^j R(s_{i+j}) - V(s_i))$ for the weight. Thus, I’m wondering if there is a detailed derivation of this objective in Eqn. 6?
- In Fig. 2, the results look quite interesting and promising. One thing I’m curious about is that SMODICE actually performs worse in some cases with less data removed. For example, SMODICE completely fails on Walker2d_1/20, 1/10, 1/5, but achieves a fairly strong performance on Walker2d_1/3. Could you provide some additional explanations on this?
- The experiments mainly focus on continuous control tasks, where the agent has repeating cyclic patterns for its actions. With this in mind, it is a bit unclear to me why incomplete trajectories would be an issue, if the dropped trajectory segments are repeating the remaining ones. In this case, does the performance gain come from the use of unlabeled data or the better Q-value approximation?

References:

[1] Peng, X. B., Kumar, A., Zhang, G., & Levine, S. (2019). Advantage-weighted regression: Simple and scalable off-policy reinforcement learning. arXiv preprint arXiv:1910.00177.


**Limitations:**

As discussed above, this paper has certain limitations regarding its presentation, novelty, the mathematical correctness, and its novelty. I do appreciate the authors’ efforts on the extensive experiments and detailed appendix. However, I would think the paper is not ready for publication yet and major revision is needed for it to be published in another venue.

---

> ### Author Rebuttal · Authors · 2023-08-09
>
> Thanks for valuable feedback.
>
> ### Q1. Inaccurate statements.
>
> - Offline imitation from observations aims to solve MDPs where only task-specific expert states and task-agnostic non-expert state-action pairs are available.
>
> This is correct. Note, we are discussing offline **imitation (Learning) from Observations (LfO)**, not **Imitation Learning (IL), i.e., no expert actions.** ValueDICE [20], DemoDICE [21] and Behavior Cloning (BC) all require expert actions missing in LfO. With only expert states and no interactions, the **Task-Agnostic (TA)** dataset is the only source for MDP dynamics.
>
> - (As mentioned in Sec. 2,) DICE methods consist of three parts: reward generation, optimization of the value function V(s), and weighted behavior cloning.
>
> **1. This sentence needs to be put in the paper’s context.** As we write right before, “As mentioned in Sec. 2”, the “DICE methods” here refers to those discussed in Sec. 2. We only mention SMODICE [1] and LobsDICE [2] in Sec. 2, and they both satisfy the statement.
>
> **2. ValueDICE requires expert actions; it is not for LfO discussed in the paper.** One can adapt ValueDICE to the LfO setting as shown by OPOLO [22]. But such a baseline is improved upon by OPOLO, which is in turn outperformed by our baseline LobsDICE.
>
> 3. To prevent confusion, we will change the sentence to “the DICE methods for offline LfO discussed above”. However, we think the message within the context has no factual error.
>
> ### Q2. Limited contribution beyond Advantage Weighted Regression (AWR) style [23] methods.
>
> **1. Our method is not a variant of AWR.** Our goal is to learn more from expert trajectories in TA datasets. We adopt the reward from SMODICE / LobsDICE and improve  with Positive-Unlabeled (PU) learning. We adopt the pessimism principle in offline RL by giving small weights to non-expert data, and use the exp function for thresholding. We sum over future returns, as the policy does not depend on the past in an MDP. The discount factor balances expert experience propagation (line 56-58) and avoidance of non-expert segments in the middle. AWR is totally different: it iteratively maximizes the (approximated) advantage from policy in the last iteration with divergence constraint.
>
> **2. The AWR methods listed in the review learns a value function, which we try to avoid.** We verify empirically that the value function is hard to obtain in our settings, e.g. missing steps in TA datasets. Also, **consistent with prior work [8], MARWIL [7] (single-iteration AWR) struggles in our settings.** See pdf in the global response for reward curves.
>
> 3. **We provide theoretical (Sec. C) and empirical evidence on the downsides of SOTA methods, SMODICE and LobsDICE,** which is another novelty.
>
> ### Q3. Stability issue of discriminator learning with coupling min-max objectives. Tricks?
>
> **1. The discriminator training is not a min-max game.** Only one moving part (c(s) or c’(s)) exists in Eq. 4 and 5 respectively. The max operator inside the min objective is a clipping technique for debiasing (see Sec. A for details).
>
> 2. Lipschitz regularizer is used for better stability (see Sec. F.4.7).
>
> 3. Optimization fails with extreme hyperparams or no Lipschitz regularizer. (see Sec. F.4.2, F.4.3 and F.4.7).
>
> ### Q4. Missing derivations for Eq. 6.
>
> As stated in Q2, our method is not a variant of AWR. Thus, Eq. 6 is immediately intuitive: the weight for BC is determined via a discounted sum of future returns; the exp function serves as a thresholding method upweighting expert data, as opposed to AWR’s closed-form solution.
>
> ### Q5.  Abnormal SMODICE-KL behavior in ablation.
>
> **1. Collapses of SMODICE-KL come from value function divergence.** As stated in Sec. C.2.1 in the appendix, SMODICE-KL diverges with missing data in the task-agnostic trajectories. When this happens (e.g., halfcheetah_1/5), SMODICE-KL collapses; otherwise, it performs well (see pdf in the global response for visualization).
>
> **2. The smoothing effect of the Neural Network (NN) mitigates the divergence, but the strength of the effect varies.** As the data distribution differs, the smoothing effect varies across datasets. Thus, the time it takes for divergence differs.
>
> **3. With more and uniform updates, the abnormality caused by NN smoothing disappears.** As shown in Fig. 19, with larger batch size, the updates on each data point are more frequent and uniform. Thus, the trend that increased noise leads to worse performance on SMODICE-KL (thin blue line) is obvious.
>
> ### Q6. Is the performance gain coming from the use of unlabeled data or better Q-estimation, as the environment has cyclic patterns?
>
> **1. Our method performs well in non-cyclic environments.** Kitchen (Sec. 4.3, 4.4), antmaze (Sec. 4.3-4.5) and pointmaze (Sec. 4.4) are non-cyclic environments; our method works well on all of them.
>
> **2. Cyclic patterns are more beneficial to baselines.** With cyclic patterns, the states are closer to each other, and NN smoothes out the diverging terms in SMODICE / LobsDICE more easily. In non-cyclic environments, our method gets better R(s) as the expert trajectories lead to a different direction from non-expert ones in the state space, but the divergence terms in SMODICE / LobsDICE remain.
>
> 3. We are unsure about what the reviewer refers to with “the use of unlabeled data”.
>
> **a) If it is the TA dataset, then it is always crucial.** As mentioned in Q1 point 1, TA dataset is the only source for MDP dynamics in offline LfO.
>
> **b) If it is PU learning that leverages the unlabeled essence of the TA dataset, then there is indeed a performance gain. In Sec. F.6**, our PU learning is much better than ORIL’s (and no PU; see pdf in the global response) on environments like halfcheetah_mismatch.
>
> **4. Better Q-estimation is important. Sec. F.6** shows that ORIL with our reward still works badly. In fact, our motivation is to avoid learning it and choose a non-parametric but robust approach.
>
> ### References
> See global response.

---

> > ### Comment · Reviewer_Re2g · 2023-08-13
> >
> > I appreciate the detailed responses by the authors. They have addressed lots of my concerns. However, I disagree with the explanation about the connection to AWR. In fact, AWR is derived from Reward-Weighted Regression (RWR) [1], which uses exactly the same form as the proposed method. However, the underlying derivation is shared. I still feel that from the theoretical side, this work is relatively weak and needs improvement. I will improve my rating to 4.
> >
> > References:
> >
> > [1] Peters, J., & Schaal, S. (2007, June). Reinforcement learning by reward-weighted regression for operational space control. In Proceedings of the 24th international conference on Machine learning (pp. 745-750).

---

> > > ### Author Response · Authors · 2023-08-15
> > > **Further Response to the Reviewer**
> > >
> > > Thanks a lot for appreciating our response and for your valuable advice!  We are glad that our response has addressed most concerns. Below are further clarifications for the remaining points:
> > >
> > > 1. Our contribution is *in the context of offline Learning from Observations (LfO).* We analyze the shortcomings of prior SOTA, and present a new, effective, yet arguably simple method. We thank the reviewer for connecting the objective of our method and RWR. However, this similarity does not hinder the novelty of our method. Note that **RWR did not invent the objective either**. The objective dates back to at least the *related payoff procedure* ([1], Sec. 11.4; [2]) which introduces an expectation-maximization (EM) procedure for RL. Since then, many works including RWR [3-6] have been built on this basis, each with their own contribution. Therefore, we think that **the novelty does not depend on using such an objective, but on how we adapt the objective to our problem setting**.
> > >
> > > 2. Based on 1, despite the resemblance of the policy objectives, the most important difference between our method and RWR is that **our method has no iterative E-step for the critic and no M-step for the actor, and cannot be derived by following RWR, or more generally EM for RL**, as EM requires sampling state-action pairs from the policy in the last iteration, which isn’t straightforward in an **offline** setting. There are two workarounds: 1) importance sampling, and 2) naively using only one iteration. The former is known to be non-robust [7] and we show in our global response that the latter (MARWIL) is ineffective. Meanwhile, non-iterative policy learning with return estimated from task-agnostic rollouts in general has been proven successful by recent RL via supervised learning works [8, 9].
> > >
> > > 3. Even within the RWR line of work, our method still has a unique contribution for our problem of interest. RWR itself is quite different from our work:
> > >
> > > a) RWR only considers immediate rewards rather than episodic returns ([10], Sec. 7). We showed the ineffectiveness in our global response pdf ($\gamma=0$).
> > >
> > > b) RWR has an adaptive reward scaling term $u_\tau(r)$ due to the EM framework. We don’t have this term. This term is not guaranteed to preserve an optimal policy and thus is not necessarily an advantage [10].
> > >
> > > Some follow-up RWR works are closer to ours. To our best knowledge, the most similar one is PoWER [5], the only work that 1) considers episodic return, 2) has no adaptive reward scaling term, and 3) uses a non-parametric approximation for the advantage or Q-value (see our previous reply for the importance of 3)). But still, three key differences remain:
> > >
> > > a) Most importantly, in our scenario, the reward label is missing. Our work improves the reward used in SMODICE and LobsDICE with Positive-Unlabeled (PU) learning. In contrast, PoWER (**and all RWR works**) assume the reward labels are available;
> > >
> > > b) PoWER uses a bilinearly parameterized policy, while we advocate for a more flexible MLP policy;
> > >
> > > c) PoWER works on a finite-horizon MDP, while we use a discount factor for future returns. Consider the kitchen environment in our paper with subtasks A, B, C and D. Assume we have a task-agnostic trajectory finishing A, B, C in order, while our sequence of interest is A, D, C. The discount factor prevents our method from blindly following the experience for B after A as it brings some degree of myopicness.
> > >
> > > Those differences are crucial for solving our task of interest, i.e., offline LfO.
> > >
> > > To sum up, we think our proposed and arguably simple method has merits that are valuable and are of interest to the LfO community. We genuinely appreciate your advice to strengthen our work; we will add this discussion to the revised version. Thanks a lot for your time and efforts. Please feel free to reach out with any additional comments that you may have.
> > >
> > > References:
> > >
> > > [1] G. E. Hinton. Connectionist Learning Procedures. 1989.
> > >
> > > [2] P. Dayan and G. E. Hinton. Using Expectation-Maximization for Reinforcement Learning. In Neural Computation, 1997.
> > >
> > > [3] J. Peters and S. Schaal. Reinforcement learning by reward-weighted regression for operational space control. In ICML, 2007.
> > >
> > > [4] J. Peters et al. Relative Entropy Policy Search. In AAAI, 2010.
> > >
> > > [5] J. Kober and J. Peter. Policy Search for Motor Primitives in Robotics. In NeurIPS, 2008.
> > >
> > > [6] T. Osa and M. Sugiyama. Hierarchical Policy Search via Return-Weighted Density Estimation. In AAAI, 2018.
> > >
> > > [7] O. Nachum et al. DualDICE: Behavior-Agnostic Estimation of Discounted Stationary Distribution Corrections. In NeurIPS, 2019.
> > >
> > > [8] L. Chen et al. Decision Transformer: Reinforcement Learning via Sequence Modeling. In ICML, 2022.
> > >
> > > [9] D. Brandforbrener et al. When does return-conditioned supervised learning work for offline reinforcement learning? In NeurIPS, 2022.
> > >
> > > [10] M. Strupl et al. Reward-Weighted Regression Converges to a Global Optimum. In AAAI, 2022.

---

> > > > ### Author Response · Authors · 2023-08-20
> > > > **Thanks for Your Valuable Advice**
> > > >
> > > > As the end of the author-reviewer discussion period approaches, we would like to thank you again for your valuable advice in the review and the response. We hope our response has addressed your remaining concerns and appreciate your feedback to our response. Please let us know if you have any further questions, and we are more than happy to clarify.

---

### Official Review · Reviewer_6nYj · 2023-07-07

**Soundness:** 3 good
**Presentation:** 3 good
**Contribution:** 3 good
**Rating:** 6
**Confidence:** 3

**Summary:**

This paper provides an offline imitation learning method upon a specific problem setting where task-specific expert states (observations) are restrictively available and task-agnostic non-expert state-actions are supplementarily available.

In this problem setting, the authors follow the line of works using DICE-based imitation learning methods, and present TAILO, Trajectory-Aware Imitation Learning from Observations. Specifically, TAILO employs the 2-step positive-unlabeled learning scheme to have the state discriminator, and then it uses rewards calculated by the discriminator as the weight for weighted behavior cloning.

This procedure in TAILO hinges on the assumption that in the pool of task-agnostic data, there exist trajectories or long segments that are closely aligned to the optimal expert trajectories for the specific target task.

The authors focus on different use cases of offline imitation learning on incomplete trajectories and show the benefits of TAILO in such context for those cases through experiments.


**Strengths:**

This paper contributes to the area of offline imitation learning, particularly in scenarios where offline data conditions exhibit variability due to incomplete trajectories. The robustness of the proposed method, Trajectory-Aware Imitation Learning from Observations (TAILO), is thoroughly demonstrated in different data conditions through the experiments detailed in Section 4.

These experimental conditions encompass scenarios with incomplete expert trajectories, incomplete task-agnostic trajectories, observations, and examples. The tests conducted appear to be comprehensive and the obtained results align well with the authors' focus, thereby solidifying the consistency and potential effectiveness of TAILO.


**Weaknesses:**

TAILO, as outlined in the paper, comprises two primary techniques, each of which is described in Section 3.2 and 3.3. One shortcoming of this work, however, is the absence of ablation studies that evaluate these proposed techniques separately.

To clarify the effectiveness of the Positive-Unlabeled (PU) learning method, alternative PU algorithms could be implemented and compared with the proposed technique. It would also be instructive to remove some losses as detailed in Equations (4) and (5) and assess the performance impact.

In a similar vein, various weighting strategies could be trialed in the context of behavior cloning, including cases where no weight is applied at all. Carrying out these ablation studies could provide a clearer understanding of the individual effectiveness and contributions of each method within the TAILO framework.

Minor errors:
- The legend of Figure 5 could be relocated without overlap.


**Questions:**

Could the authors compare the recent work [1] in offline imitation learning?
[1] Discriminator-Weighted Offline Imitation Learning from Suboptimal Demonstrations (ICML 2022)

Could the authors introduce some specific real-world application scenarios for the problem settings of incomplete trajectories?

Is TAILO the first that applies PU in the context of offline imitation learning? Could the authors further clarify the novelty of their work?



**Limitations:**

The limitation is described in the conclusion section.

---

> ### Author Rebuttal · Authors · 2023-08-09
>
> Thanks for valuable feedback.
>
> ### Q1. The absence of ablation studies in Sec. 3.2 and 3.3.
>
> Part of the requested ablation studies are in **Sec. F.6.** We test the remaining ablations and summarize here.
>
> **1. The ablation of Sec. 3.2.** In Sec. F.6, we compare ours and ours-V2, where we study the effect of using the ORIL-style Positive-Unlabeled (PU) learning technique instead of ours. As Fig. 23 and 24 suggest, there are significant differences in the Antmaze and halfcheetah_mismatch environment and marginal differences in the kitchen environment. We additionally test using no PU learning at all, which fails on halfcheetah_mismatch and hopper_head50_tail50 environments (see pdf in the global response for reward curves).
>
> **2. The ablation of Sec. 3.3.** In Sec. F.6, we compare ours and ORIL-logR-V2, and ours and ours-V1. We find that the non-parametric retrieval of weights for Behavior Cloning (BC) is much better than the RL process introduced by ORIL, while ORIL is non-robust and often diverges. We also find that the log design of $R(s)=\log\frac{c(s)}{1-c(s)}$ is much better than the linear design $10c(s)$, as the former further emphasizes the reward given by states that are clearly classified as expert states.
>
>
> ### Q2. Alternative Positive-Unlabeled (PU) learning algorithms and removal of some losses in Eq. 4 and 5.
>
> **The requested ablation studies can be found in point 1 in Q1.** As stated in Q1, the result shows that our design of PU learning exhibits a significant improvement on certain environments like halfcheetah_mismatch and hopper_head50_tail50.
>
> ### Q3. Ablations on BC weighting strategies, including unweighted strategy.
> We test three different weighting strategies in the main paper and the appendix. We additionally test another two weighting strategies and summarize here (see pdf in the global response for reward curves).
>
> 1. We tested SMODICE [1], LobsDICE [2], ORIL [3] and ReCOIL [9] (**Sec. F.2**) as representatives of value-based weights (the dual variable of DICE methods can be seen as an equivalent of the value function).
>
> 2. We use plain, unweighted behavior cloning as a standard baseline in all experiments.
>
> 3. In **Sec. F.6** in the appendix, we test Ours-V1 to see whether the design of $R(s)=\log\frac{c(s)}{1-c(s)}$ instead of being proportional to the discriminator output is reasonable.
>
> 4. We additionally test OTR [10] as the representative of Wasserstein-based weights, which does not work well in case of small portions of expert data in the task-agnostic data.
>
> 5. We additionally test MARWIL [7] in the global response as the representative of advantage-based weights. It performs similarly to plain BC. This is consistent with prior work [8].
>
> **We found TAILO to significantly outperform all five strategies.**
>
> ### Q4. Minor Errors on the legend of Fig. 5.
>
> Thanks for the advice. As this year’s NeurIPS does not allow modification of the original draft, we will modify the figure in the camera-ready revision.
>
> ### Q5. Comparison to DWBC [16]
>
> **1. DWBC is not applicable to offline Learning from Observations (LfO).** In DWBC, the input of the discriminator takes state, action and $\log\pi(a|s)$ from the task-specific and task-agnostic data as input, the latter two of which are not computable in LfO as expert actions aren’t available. Further, DWBC uses three terms in its policy loss (see Eq. 5 of DWBC), but we are unable to calculate the first two terms due to the lack of expert actions in LfO data. The adaptation of DWBC to offline LfO is beyond the scope of this work.
>
> **2. Even with extra access to expert action, DWBC is still unable to learn well from task-agnostic data with a very small portion of expert trajectories.** See pdf in the global response for reward curves.
>
>
> ### Q6. Real-World Applications for Incomplete Trajectories
>
> There are at least two possible applications for incomplete trajectories:
>
> **1. Learning from corrupted task-agnostic data.** In real-world scenarios, task-agnostic data are often accumulated from many different experiences, possibly with recording devices turned on and off in the middle, or even from the wild with unfamiliar sources and different formats (in which case alignment is required [17]). Therefore, it is very likely that some of the data are corrupted with a few steps missing in the middle of a long trajectory.
>
> **2. Learning from expert key frames and goals.** In robotic tasks such as navigation, it is common that a robot needs to move to another point with a few waypoints or the goal location as clues [18]; in physical simulation, we need to generate the desired agent behavior [19] by a few frames designated by the artists, as manually determining the position of every frame is prohibitively labor-intensive. In both cases, the task-specific dataset is incomplete with only examples or a few key frames.
>
> ### Q7. The novelty of using Positive-Unlabeled (PU) learning.
>
> 1. TAILO is not the first work to apply PU learning in offline Imitation Learning (IL); there is one prior work, ORIL [3], that also applies PU learning in offline IL. However, the two methods are significantly different; a thorough ablation in **Sec. F.6** shows that each difference between TAILO and ORIL leads to our performance gains.
>
> 2. The novelty of TAILO can be summarized as follows:
>
> a) a novel offline imitation from observation method that works well on a variety of problems;
>
> b) a simple and non-parametric way of determining weights for weighted behavior cloning;
>
> c) a novel way of acquiring a discriminator-based reward, inspired by positive-unlabeled learning designs from prior work, which is empirically shown to be effective.
>
> d) both theoretical (**Sec. C**) and empirical analysis on the shortcomings of state-of-the-art SMODICE and LobsDICE.
>
> ### References
> See global response.

---

> > ### Comment · Reviewer_6nYj · 2023-08-20
> >
> > I would like to extend my thanks for the comprehensive response, which addresses most of the concerns I raised, particularly regarding the ablation studies. I maintain my original score of weak accept, as my viewpoint on the novelty remains largely unchanged.

---

### Official Review · Reviewer_d8U7 · 2023-07-07

**Soundness:** 3 good
**Presentation:** 3 good
**Contribution:** 2 fair
**Rating:** 5
**Confidence:** 3

**Summary:**

This paper studies offline imitation from observations, assuming a small amount of task-specific expert states and task-agnostic non-expert state-action pairs are available. The method is to learn a discriminator to identify expert states in the task-agnostic dataset and then apply weighted behavior cloning to imitate states. Empirical results show that the proposed method TAILO outperforms state-of-the-art methods (the family of DICE), particular for datasets with incomplete trajectories.

**Strengths:**

1) The idea behind this approach is intuitive, well-motivated, and conceptually simple.
2) Empirically, TAILO performs well in learning from datasets with incomplete trajectories, significantly better than state-of-the-art approaches.
3) The empirical evaluation is extensive. Considering different ways and hyper-parameters to modify datasets, the authors show good performance across all these datasets. It is impressive that the same set of hyper-parameters for TAILO performs well in all these experiments.


**Weaknesses:**

1) Compared with DICE methods (and the previous state-of-the-art), the proposed method lacks a theoretical foundation. The objective functions in equations 4,5,6 look mathematically complex, but we do not have a theory to support them.
2) The ablation study is missing. Thus, the importance of each component in TAILO is unclear. See the section of Questions for more details.
3) In experiments, the datasets are mostly manipulated, different from the original dataset used in previous work. This seems not standard for benchmarking and comparing approaches.

**Questions:**

1) About the role of c'(s) in Section 3.2, if we directly learn c(s) without the help of c'(s), how much it will hurt the performance of TAILO?
2) In lines 150-152, intuitively, the value of hyper-parameter \beta_1 should highly affect the final performance. Could you show an ablative study on this? Do you have any intuitive explanation why the same \beta_1 works well across different experiments (as mentioned in line 194)?
3) Could you please show experimental results on datasets exactly the same as the SMODICE paper? On these modified datasets, the performance of SMODICE looks poor. It either always keeps a low score or suddenly crashes. I'm wondering about the performance of TAILO on the standard benchmark datasets. It is great to clarify, given a specific dataset, how should we choose between SMODICE and TAILO? Is TAILO always the better choice?

**Limitations:**

This work is based on the assumption that there exist expert trajectories/segments for the task of interest in the task-agnostic dataset. I'm concerned about the reliability or generalizability of this assumption. In real-world problems, when collecting demonstration dataset, there seems a small chance that this assumption holds.

---

> ### Author Rebuttal · Authors · 2023-08-09
>
> Thanks for valuable feedback.
>
> ### Q1: Lack of theoretical foundation, especially for Eq. 4, 5 and 6.
>
> **1. Eq. 4 and 5 do not lack theoretical foundations.** The objective functions in Eq. 4 and 5 come from established Positive-Unlabeled (PU) learning works [24]. Eq. 4 is a binary classification objective that estimates the loss from the negative data using a mixed, unlabeled data distribution, with a max operator that limits biased estimates; Eq. 5 is a selection between Eq. 4 and a standard binary classification loss. See **Sec. A** in the appendix (starting from line 509) for a detailed derivation of Eq. 4 and 5.
>
> **2. Eq. 6 is immediately intuitive.** Eq. 6 states that the weight for Behavior Cloning (BC) is determined by the discounted future return in the trajectory from the current state; the exp function serves as a thresholding method to emphasize the importance of trajectories with high return.
>
> **3. TAILO is based on intuitive improvements of some carefully studied DICE methods and it is effective.** The idea of TAILO is that we want to follow the reward function design used in SMODICE and LobsDICE, but we do not want their non-robust weight update (see **Sec. C** for the reasons of non-robustness). Instead, we find that a weight as simple as thresholded future reward along the trajectory works better. Furthermore, we want to improve the classifier training of SMODICE and LobsDICE. As some examples are unlabeled, PU learning is introduced.
>
> ### Q2: Ablation studies are missing.
>
> All ablations mentioned in the question section were already provided in the appendix.
>
> **1. The role of c’(s).** The result of directly learning c(s) without the help of c’(s) is the “ours-V2” variant in **Sec. F.6** (see Tab. 4 in the appendix for the meaning of each variant). As Fig. 23 suggests, there is a significant gain in the Antmaze environment and a marginal gain in the kitchen environment with c’(s).
>
> **2. The effect of $\beta_1$.** The ablation study is in **Sec. F.4.2**. According to Fig. 14, The reward will only decrease with extreme selection of $\beta_1$. Intuitively, there are two reasons why the same $\beta_1$ works well for different experiments: 1) the ratio of expert trajectories in our task-agnostic dataset is small (<5%), thus there is little expert data erroneously classified as safe negatives; 2) the Lipschitz-smoothed discriminator yields a good classification margin, and thus the 60%+ trajectories with lower average reward represents all non-expert data well.
>
> **3. Experimental results on datasets identical to those in the SMODICE paper.** The result is in **Sec. F.5.** As Fig. 22 suggests, our method is still better. Note SMODICE works well in their paper because they report the better result among using KL and chi-squared divergence (and in Sec. E.2. of the SMODICE appendix one can see the divergence chosen is crucial), while we report them separately. On all the datasets that SMODICE was tested on, TAILO improves, as TAILO is at least comparable to the better variant of SMODICE, while removing the need of selecting a divergence.
>
> ### Q3. Using manipulated dataset from prior work is not standard for benchmarking.
>
> **1. We also improve on standard benchmarks.** As stated in Q2 point 3, our method also improves on standard benchmarks. We also compare to offline RL methods on standard D4RL datasets (Q2 of reviewer iGFb), where the other methods have extra access to ground-truth reward labels. Nonetheless, the proposed approach performs well.
>
> **2. The original benchmark is close to being solved and cannot reflect the challenges discussed in our work, e.g., incomplete trajectories.** Thus, we construct new benchmarks to show that our method has even larger benefits against those challenges, and We think that researchers should go beyond the standard benchmarks.
>
> **3. Our comparison is fair as our testbed covers that of SMODICE’s.** SMODICE tests standard imitation from observation (Sec. F.5 in our work), imitation from examples (Sec. 4.4), and learning from mismatched dynamics (Sec. 4.5). We test TAILO on **every environment in every setting with exactly the same condition** and find consistent, significant gains. We further prove that our method is robust to incomplete trajectories (Sec. 4.2, 4.3) and task-agnostic datasets with fewer expert data (Sec. 4.1).
>
> **4. Our modification for demonstration of advantages is minimal.** Except for mismatched dynamics where we strictly follow SMODICE, we do not make any modification to the environment itself; we only modify the dataset.
>
> ### Q4. The assumption that the task-agnostic dataset contains expert segments does not hold in real-world problems.
>
> **1. The assumption is common in prior works.** This assumption is the basis of prior works in the skill-based learning community, e.g., SPiRL [12], SKiLD [13] and FIST [14], where they assume that there are many action sequences from the task-agnostic data that can be utilized for the task of interest. Practically, the benchmarks of many recent works in offline imitation learning satisfy the property in the assumption (note mujoco medium and medium-replay data also contain expert trajectories, as shown in Fig. 4 of OTR [10] and Fig. 4 of decision transformer [6]), such as SMODICE [1], LobsDICE [2], OTR [10], ReCOIL [9], MAHALO [15], and decision transformer [6].
>
> **2. The assumption has real-world applications.** For example, in the robotics community, the robot often needs to utilize overlapping skills, i.e., trajectory segments observed in other tasks, such as moving the robotic arm to a particular place and grabbing the item, to complete the current task. This is demonstrated in our kitchen environment, where the robotic arm needs to combine different skills (e.g. opening the cabinet, moving the kettle) to complete a particular procedure of interest.
>
> ### References
> See global response.

---

> > ### Comment · Reviewer_d8U7 · 2023-08-15
> > **Thanks for the detailed response**
> >
> > Thank you for the detailed response to resolve my concerns.  About the theory, it will be more clear to include such an explanation in the main text to help understand the equations. However, the theory behind Equations 4,5,6 was all from previous work, which is not the contribution of this submission. It is great to see more extensive experiments in the appendix about the ablation study. The response resolves most of my problem, but the contributions are not strong enough (especially the theoretical part). So I change the rating to just slightly lean toward acceptance.

---

> > > ### Author Response · Authors · 2023-08-17
> > > **Further Response to Reviewer d8U7**
> > >
> > > Thanks a lot for appreciating our response and for your valuable advice! We will follow your suggestion and include the explanation in the main text. Below are further clarifications to the reviewer’s response:
> > >
> > > **Q1. The theory behind Eq. 4, 5, 6 was all from prior work and thus is not the contribution.**
> > >
> > > While Eq. 4 is from prior work, Eq. 5 is a new and important combination of Eq. 4 and binary classification. More critically, our two-step training paradigm of positive-unlabeled (PU) learning that utilizes Eq. 4 and 5 is novel for offline imitation learning. We showed its success in the appendix **Sec. F.6**: directly using Eq. 4 does not work on environments like halfcheetah-mismatch, while our two-step PU learning works well (normalized reward of 60 (ours) vs 0 (one-step PU); illustrated in **Fig. 6**).
> > >
> > > Also, in our latest response to Reviewer Re2g, we discussed in detail the unique contribution of our work to the RWR [1] line of works which use an objective similar to Eq. 6. Below is a brief summary of the differences:
> > >
> > > 1. Prior RWR works that use the policy objective of Eq. 6 assume reward labels are available, which differs from our work that combines the objective with reward from PU learning.
> > >
> > > 2. Prior RWR works that use the policy objective of Eq. 6 are theoretically derived from the Expectation-Maximization (EM) framework, while our work isn’t. Straightforward adaptation, such as MARWIL, is shown to be ineffective for our setting.
> > >
> > > 3. Our design choices, e.g., non-parametric return estimation, differs from each of the prior RWR works.
> > >
> > > **Q2. The theoretical contribution is not strong enough.**
> > >
> > > We believe that our contribution is significant for offline Learning from Observations (LfO), because we 1) analyze *both theoretically and empirically* the shortcomings of prior SOTA in LfO, and 2) propose a solution that is new, effective, yet arguably simple and leads to empirical success in a variety of scenarios. We believe that the significance of our contributions in LfO should not be underestimated, due to the simplicity and resemblance of the formulation of our solution to prior works proposed in different contexts or tasks.
> > >
> > > References:
> > >
> > > [1] J. Peters and S. Schaal. Reinforcement learning by reward-weighted regression for operational space control. In ICML, 2007.

---

### Official Review · Reviewer_rs8a · 2023-07-07

**Soundness:** 3 good
**Presentation:** 4 excellent
**Contribution:** 3 good
**Rating:** 6
**Confidence:** 3

**Summary:**

The authors propose TAILO, Trajectory-Aware Imitation Learning from Observations, a method to solve MDPs from offline data in the form of task-specific expert states and task-agnostic state-action trajectories. The method addresses the instabilities of algorithms like DICE, takes the context of a trajectory into account even when it may be incomplete, and demonstrate across many Mujoco environments that TAILO outperforms baselines especially with incomplete trajectories.

**Strengths:**

+ [Clarity] The paper is well-written, organized, and easy to follow.
+ [Clarity] The authors include relevant background about DICE and document baselines clearly.
+ [Originality] The authors propose a relatively simple solution, which is novel to the best of my knowledge.
+ [Quality] Experiments on several Mujoco tasks demonstrate compelling performance. The authors additionally report results on the Franka kitchen environment.

**Weaknesses:**

-

**Questions:**

-

**Limitations:**

-

---

> ### Author Rebuttal · Authors · 2023-08-09
>
> We thank the reviewer for appreciating our work and are excited about the positive feedback. Thanks a lot.

---

### Official Review · Reviewer_iGFb · 2023-07-10

**Soundness:** 3 good
**Presentation:** 3 good
**Contribution:** 2 fair
**Rating:** 5
**Confidence:** 4

**Summary:**

This paper proposes a simple weighted BC algorithm for offline RL with missing data. The authors first study that DICE family of algorithms suffer from missing data due to inaccurate value estimation or sparsity of observations which causes undesired monotonicity. The authors propose training a reward model from PU data and use MC value estimation with BC to train a policy. Experimental results on offline RL benchmarks show that this method is more robust to missing data and outperforms previous DICE family of methods.

**Strengths:**

The paper discusses limitations of DICE family of models in a practical way. The proposed method is also practical in a real setting. Results show robustness and overall improvement in missing/noisy data regime.

**Weaknesses:**

- The paper studies DICE family of algorithms with noisy data but the proposed method is mainly independent of these findings. It is a new objective without any value function or duality, which feels like it is not focusing on circumventing issues with DICE.

- Model based methods are missing from the comparison.

-  The proposed method is similar to [1] in which the authors use exponential of advantage with BC for batch RL setting. While they don’t study missing data setting, I believe the objective is more general and can be applied to yours as well. A detailed comparison would be helpful.

- Which one is more crucial: learning a better reward using PU data or using discounted future rewards for the value estimate? What is the performance if you used one-level PU learning without safe examples or two-level PU learning with 1-step reward?

- Eq (6) doesn’t handle missing data explicitly. What happens if some $i+k,k>0$ is missing from the trajectory? Do you just ignore that from the summation?

- I would have expected that with more noise, DICE methods perform worse. But in Figure-2, SMODICE-KL performs well Walker2d_1/3 and worse with Halfcheetah_1/5 while for others it is the opposite. Could you please explain why?





[1] Exponentially Weighted Imitation Learning for Batched Historical Data. Qing Wang, Jiechao Xiong, Lei Han, peng sun, Han Liu, Tong Zhang.

**Questions:**

Please see above for specific questions.

**Limitations:**

The authors study some limitations but a discussion on whether progress on D4RL reflects progress on real life would be helpful.

---

> ### Author Rebuttal · Authors · 2023-08-09
>
> Thanks a lot for the valuable feedback.
>
> ### Q1. The paper studies DICE with noisy data, but the proposed method is not focusing on circumventing issues with DICE methods.
>
> Though our method looks very different, our motivation is to solve the issues in DICE methods as simply as possible. Below is a summary of our motivations:
>
> **1. We want to adopt the reward design of SMODICE [1] and LobsDICE [2], as they are empirically effective.** However, with expert segments in the task-agnostic dataset, the reward design with normal binary classification objective sometimes fails. For this reason we improve by introducing Positive-Unlabeled (PU) learning.
>
> **2. We want to address the problem of SMODICE and LobsDICE that they require complete task-agnostic trajectories to work well.** We found theoretically (**Sec. C** in the appendix) that SMODICE and LobsDICE formulations are brittle if incomplete trajectories are present. We also observed this empirically as the value function update is often non-robust (which is also true for ORIL [3]).
>
> Based on this, we find that simply using the discounted sum of future exp(R(s)) is a good solution to address both points and improve on all testbeds of SMODICE. Thus, though the final methods are very different, our idea is in spirit an improvement over DICE methods.
>
> ### Q2. Comparison to model-based methods.
>
> Great suggestion. We additionally compare our method to two model-based offline RL methods MOReL [4] and MOPO [5]. Notably, MOReL is much slower to train, requiring 2 days as opposed to 4 hours of our approach on our machine (NVIDIA RTX 2080Ti). To improve efficiency, we compare our method to MOReL and MOPO which are provided with ground-truth reward labels, i.e., MOReL and MOPO have an advantage. The trajectory with the best return is provided as the task-specific dataset to our method. Despite being agnostic to reward labels, our method is still **marginally better than MOReL** (**74.3 vs. 72.9** reward averaged over 9 environments {halfcheetah, hopper, walker2d} * {medium, medium-replay, medium-expert}), and **much better than MOPO** (**74.3 vs. 42.1** average reward).
>
> Below is the detailed performance comparison:
>
> | Dataset | MOReL | MOPO | TAILO (ours) |
> |---|---|---|---|
> |halfcheetah-Medium(M)|42.1|**42.3**|39.8|
> |hopper-M|**95.4**|28|56.2|
> |walker2d-M|**77.8**|17.8|71.7|
> |halfcheetah-Medium-Replay(MR)|40.2|**53.1**|42.8|
> |hopper-MR|**93.6**|67.5|83.4|
> |walker2d-MR|49.8|39.0|**61.2**|
> |halfcheetah-Medium-Expert(ME)|53.3|63.3|**94.3**|
> |hopper-ME|108.7|23.7|**111.5**|
> |walker2d-ME|95.6|44.6|**108.2**|
> |average|72.9|42.1|**74.3**|
>
> ### Q3. Add comparison to MARWIL [7].
>
> We additionally compare to MARWIL as requested, and we find that consistent with prior work [8], **MARWIL performs similar to plain BC and worse than our method.** See the pdf in the global response for reward curves.
>
> ### Q4. The effect of using PU learning vs. using future reward for value estimates, and more ablations on PU learning.
>
> Part of the answer can be found in the appendix **Sec. F.6**; we additionally conduct experiments for the remaining questions and summarize here (see the pdf of the global response for reward curves).
>
> **1. Using future rewards for value estimates is more crucial, but one-level PU without safe examples also leads to a performance drop in some environments.** The comparison between ours and ORIL variants in **Sec. F.6** shows that even with rewards from PU learning, RL often fails. Using 1-step PU learning without safe examples causes a significant performance drop in some environments, as Fig. 23 and 24 depict.
>
> **2. We additionally test the result with 2-step PU learning + 1-step reward and without PU learning.** The result shows that while performance does drop without PU learning, it is less important than using the discounted sum of future reward instead of 1-step reward (i.e., $\gamma=0$) as value estimation.
>
> ### Q5. About handling missing data in the task-agnostic dataset.
>
> Yes, we ignore missing data from the summation in Eq. 6. The idea is that the weighted sum does not need to be very accurate to work well; e.g., if every other step is missing, the result will be summing over every other future state with $\gamma’=\gamma^{0.5}$. Because $\gamma$ is close to $1$ in long-horizon environments, the result does not change much.
>
> ### Q6. The unintuitive behavior of SMODICE-KL.
>
> **1. Collapses of SMODICE-KL come from value function divergence.** As stated in **Sec. C.2.1** in the appendix, SMODICE-KL diverges with missing data in the middle of the task-agnostic trajectories. When this happens (e.g., halfcheetah_1/5, walker2d_1/5 in Fig. 2), SMODICE-KL collapses; otherwise, it performs decently well (see pdf in the global response for visualization).
>
> **2. The smoothing effect of the Neural Network (NN) mitigates the divergence, but the strength of the effect varies.** As the data distribution differs, the smoothing effect varies across datasets. Thus, the time it takes for divergence differs.
>
> **3. With more and uniform updates, the abnormality caused by NN smoothing disappears.** As shown in **Fig. 19**, when the batch size is larger, the updates on each data point are more frequent and uniform. With such a batch size, the trend that increased noise leads to worse performance on SMODICE-KL (thin blue line) is more obvious.
>
> ### Q7. Limitations on the gap between D4RL and real-life progress.
>
> Great advice. We will add the following discussion to the limitation section in the camera-ready version:
>
> A limitation of our work is that our experiments are based on simulated environments such as D4RL. Thus a gap between our work and real-life progress remains. While we are following the settings of many recent works, such as SMODICE [1], ReCOIL [9] and OTR [10], to bridge the gap using techniques such as sim2real in the robotics community [11] is another very important direction for future work.
>
> ### References
> See global response.

---

> > ### Author Response · Authors · 2023-08-20
> > **Thanks for Your Valuable Advice**
> >
> > As the end of the author-reviewer discussion period approaches, we would like to thank you again for your valuable advice in the review. We hope our rebuttal has addressed your concerns and appreciate your feedback to our response. Please let us know if you have any further questions, and we are more than happy to clarify.

---

> > ### Comment · Reviewer_iGFb · 2023-08-21
> > **Thank you for your response!**
> >
> > Thank you for the clarifications and additional experimental results. I increased my score.

---

> > > ### Author Response · Authors · 2023-08-22
> > > **Response**
> > >
> > > Thanks for your appreciation of our response!

---

### Author Rebuttal · Authors · 2023-08-09

We genuinely thank the reviewers for their valuable opinions and advice. We are delighted to see that the reviewers have carefully evaluated our work, given many valuable feedbacks, and highlight that

1) we study the DICE family of algorithms in offline LfO, pointing out that they suffer from missing data due to inaccurate value estimation or sparsity of observations which causes undesired monotonicity (cited from reviewer iGFb);

2) the solution proposed in our paper is simple and effective, and

3) we have extensive empirical evaluations and a detailed appendix, which thoroughly demonstrate that our method improves upon baselines in many different scenarios.

We respond to questions in the individual replies. We also include a pdf with additional figures. Finally, we use the following references in our replies:

[1]  Y. J. Ma et al. Versatile Offline Imitation from Observations and Examples via Regularized State-Occupancy Matching. In ICML, 2022.

[2] G. H. Kim. et al. LobsDICE: Offline Learning from Observation via Stationary Distribution Correction Estimation. In NeurIPS, 2022.

[3] K. Zolna et al. Offline Learning from Demonstrations and Unlabeled Experience. In Offline Reinforcement Learning Workshop at NeurIPS, 2020.

[4] R. Kidambi et al. MOReL: Model-based Offline Reinforcement Learning. In NeurIPS, 2020.

[5] T. Yu et al. MOPO: Model-based Offline Policy Optimization. In NeurIPS, 2020.

[6] L. Chen et al. Decision Transformer: Reinforcement Learning via Sequence Modeling. In ICML, 2022.

[7] Q. Wang et al. Exponentially Weighted Imitation Learning for Batched Historical Data. In NeurIPS, 2018.

[8] X. Chen et al. BAIL: Best-Action Imitation Learning for Batch Deep Reinforcement Learning. In NeurIPS, 2020.

[9] H. S. Sikchi et al. Imitation from Arbitrary Experience: A Dual Unification of Reinforcement and Imitation Learning Methods. In ArXiv, 2023.

[10] Y. Luo et al. Optimal Transport for Offline Imitation Learning. In ICLR, 2023.

[11] S. Höfer et al. Sim2Real in Robotics and Automation: Applications and Challenges. in IEEE Transactions on Automation Science and Engineering, 2021.

[12] K. Pertsch et al. Accelerating Reinforcement Learning with Learned Skill Priors. In CoRL, 2020.

[13] K. Pertsch et al. Demonstration-guided reinforcement learning with learned skills. In CoRL, 2021.

[14] K. Hakhamaneshi et al. Hierarchical few-shot imitation with skill transition models. In ICLR, 2022.

[15] A. Li et al. MAHALO: Unifying Offline Reinforcement Learning and Imitation Learning from Observations. In ICML, 2023.

[16] H. Xu et al. Discriminator-Weighted Offline Imitation Learning from Suboptimal Demonstrations. In ICML, 2022.

[17] M. Chang et al. Semantic Visual Navigation by Watching YouTube Videos. In NeurIPS, 2020.

[18] D. S. Chaplot et al. Neural Topological SLAM for Visual Navigation. In CVPR, 2020.

[19] X. Peng et al. DeepMimic: Example-Guided Deep Reinforcement Learning of Physics-Based Character Skills. In ACM Transactions on Graphics, 2018.

[20] I. Kostrikov et al. Imitation learning via off-policy distribution matching. In International Conference on Learning Representations. In ICLR, 2020.

[21] G. H. Kim et al. Demodice: Offline imitation learning with supplementary imperfect demonstrations. In ICLR, 2022.

[22] Z. Zhu et al. Off-Policy Imitation Learning from Observations. In NeurIPS, 2020.

[23] X. B. Peng et al. Advantage-Weighted Regression: Simple and Scalable Off-Policy Reinforcement Learning. In ArXiv, 2019.

[24] R. Kiryo et al. Positive-unlabeled learning with non-negative risk estimator. In NIPS, 2017.

---

### Decision · Program_Chairs · 2023-09-21

**Decision:**

Accept (poster)

**Comment:**

The submission here identifies an important shortcoming of the DICE family of methods applied to imitation from observation and proposes an simple but interesting method for addressing this shortcoming that should be of interest to the NeurIPS community. The authors have engaged in a productive conversation with the reviewers during the discussion phase and the submission here should now be revised in order to provide the same additional results and context that was provided to the reviewers (eg, the additional ablation study, the clarity regarding motivation, and the positioning with respect to related work).